# A dynamic clamp protocol to artificially modify cell capacitance

Paul Pfeiffer[1,2], Federico José Barreda Tomás[2,3], Jiameng Wu[2,4], Jan-Hendrik Schleimer[1,2], Imre Vida[2,3], Susanne Schreiber[1,2]*

[1]Institute for Theoretical Biology, Humboldt-Universität zu Berlin, Berlin, Germany; [2]Bernstein Center for Computational Neuroscience, Humboldt-Universität zu Berlin, Berlin, Germany; [3]Institute for Integrative Neuroanatomy, Charité - Universitätsmedizin Berlin, Corporate Member of Freie Universität Berlin, Humboldt-Universität zu Berlin, and Berlin Institute of Health, Berlin, Germany; [4]Einstein Center for Neurosciences Berlin, Charité - Universitätsmedizin Berlin, Corporate Member of Freie Universität Berlin, Humboldt-Universität zu Berlin, and Berlin Institute of Health, Berlin, Germany

**Abstract** Dynamics of excitable cells and networks depend on the membrane time constant, set by membrane resistance and capacitance. Whereas pharmacological and genetic manipulations of ionic conductances of excitable membranes are routine in electrophysiology, experimental control over capacitance remains a challenge. Here, we present capacitance clamp, an approach that allows electrophysiologists to mimic a modified capacitance in biological neurons via an unconventional application of the dynamic clamp technique. We first demonstrate the feasibility to quantitatively modulate capacitance in a mathematical neuron model and then confirm the functionality of capacitance clamp in in vitro experiments in granule cells of rodent dentate gyrus with up to threefold virtual capacitance changes. Clamping of capacitance thus constitutes a novel technique to probe and decipher mechanisms of neuronal signaling in ways that were so far inaccessible to experimental electrophysiology.

## Editor's evaluation

The manuscript introduces a new enhancement to the dynamic clamp technique, CapClamp that, analogous to the artificial conductances of standard Dynamic Clamp, allows the experimenter to adjust the somatic time constant by setting a new membrane artificial capacitance independent of any change in input resistance. The technique is shown to have application for studying temporal integration, energetic costs of spiking and bifurcations. The technique is rigorously tested in model and physiological application and is robust when sampling frequency of the feedback (clamp) loop is fast compared to the fastest electrical event in a neuron (usually action potentials), and for vertebrate neurons it should be 20KHz or faster and yet faster for fast spiking neurons.

*For correspondence:
s.schreiber@hu-berlin.de

## Introduction

Membrane capacitance is a major biophysical parameter in neurons and other excitable cells, which determines how fast the membrane potential changes in response to a current (*Golowasch et al., 2009*; *White and Hooper, 2013*). How capacitance impacts electrical signaling and neuronal processing, however, can rarely be observed experimentally, because besides reduced values in myelinated axons (*Hartline and Colman, 2007*) most membranes appear to have a specific membrane capacitance in the range of 0.7–1.0 µF/cm2 (*Gentet et al., 2000*). The effects of capacitance changes

can, therefore, so far only be compared via mathematical simulations, where capacitance is simple to control. Such modeling, for example, suggests that the reduced membrane capacitance observed in human pyramidal cells can serve to increase synaptic efficacy or propagation speed of action potentials (*Eyal et al., 2016*, but see *Beaulieu-Laroche et al., 2018*). In contrast, experimental manipulation of capacitance remains challenging; in particular because changes in membrane area, thickness and lipid composition that affect capacitance might influence other membrane functions, such as the embedding of ion channels, with potentially unintended and uncontrolled consequences for electrical behavior. Here, we address this technical challenge by introducing capacitance clamp (CapClamp): an intracellular recording mode based on the dynamic clamp that emulates altered capacitance values in biological neurons (*Robinson, 1994*; *Sharp et al., 1993*). Via CapClamp, the voltage dynamics governed by the actual biophysics of a cell – active ion channels and synaptic inputs – can thus be flexibly probed under multiple 'virtual' capacitance conditions, which provides precise experimental control over this hitherto inaccessible parameter.

In addition to the analysis of biological capacitance adaptations, control over capacitance offers a distinct way to probe cellular electrical dynamics. Capacitance has a unique temporal role, because its direct effects are restricted to the membrane time constant whilst leaving the steady state I-V function unaltered. In this way, capacitance differs from leak conductance, the other determinant of the time constant, as the latter also alters steady-state response amplitudes. For this reason, theoretical studies preferentially vary capacitance to investigate ion channel dynamics (*Jaffe and Brenner, 2018*; *Franci et al., 2018*) and qualitative switches (bifurcations) in neural excitability (*Kirst et al., 2015*; *Hesse et al., 2017*). Furthermore, effects of an altered capacitance can be informative about more complex, time scale-related parameters like temperature or ion concentrations (*Contreras et al., 2020*). Such computational predictions, however, often rely on simplified neuron models, so a similar experimental control over capacitance would be desirable to test them in biological cells.

The proposed CapClamp alters capacitance in a virtual manner, combining the simplicity of computational control with the complex biophysics of a real neuron. It is inspired by the dynamic clamp technique, which has originally been developed to simulate the presence of additional conductances in a biological neuron relying on a fast feedback loop between intracellular recording and a computational model (*Robinson, 1994*; *Sharp et al., 1993*; *Prinz et al., 2004*; *Economo et al., 2010*). The precise control over these virtual conductances enables electrophysiological experiments that are more difficult or even impossible with traditional pharmacological or genetic means (*Turrigiano et al., 1996*; *Svirskis et al., 2004*; *Prescott et al., 2008b*; *Hasenstaub et al., 2010*; *Szűcs et al., 2017*; *Pfeiffer et al., 2020*). Here, we demonstrate how the dynamic clamp can be extended to enable manipulations of the apparent membrane capacitance by currents designed to speed up or slow down dynamics of the membrane potential. We derive a simple expression for these CapClamp currents, which can be applied in all excitable cells and only requires the experimenter to specify the original cell and the desired target capacitance. In an experiment based on a hardware-implemented RC circuit, we verify that the CapClamp indeed correctly modifies the time constant. Via numerical simulations, we confirm that a clamped model neuron exhibits the same pronounced changes of firing and spike shape as a control cell with an altered capacitance. For an experimental demonstration, we clamp the near-somatic capacitance of rat dentate gyrus granule cells and analyze how the induced local capacitance change affects their spiking behavior. Finally, we illustrate how the CapClamp can be used to probe signal integration and energy consumption of excitable cells in ways that so far were experimentally inaccessible.

## Results

### Capacitance clamp: A dynamic clamp protocol to mimic capacitance changes

Dynamic clamp relies on a fast feedback loop between an intracellular recording of a neuron and a computer that simulates virtual cellular or circuit components online. Originally, the dynamic clamp has been developed to study how a membrane conductance alters the neuron's voltage dynamics (*Sharp et al., 1993*; *Robinson, 1994*). In each sampling interval (i.e. time interval between two voltage samplings), a digital model of the conductance receives the sampled membrane potential, updates the conductance state and sends the corresponding current value back to the amplifier. Given a

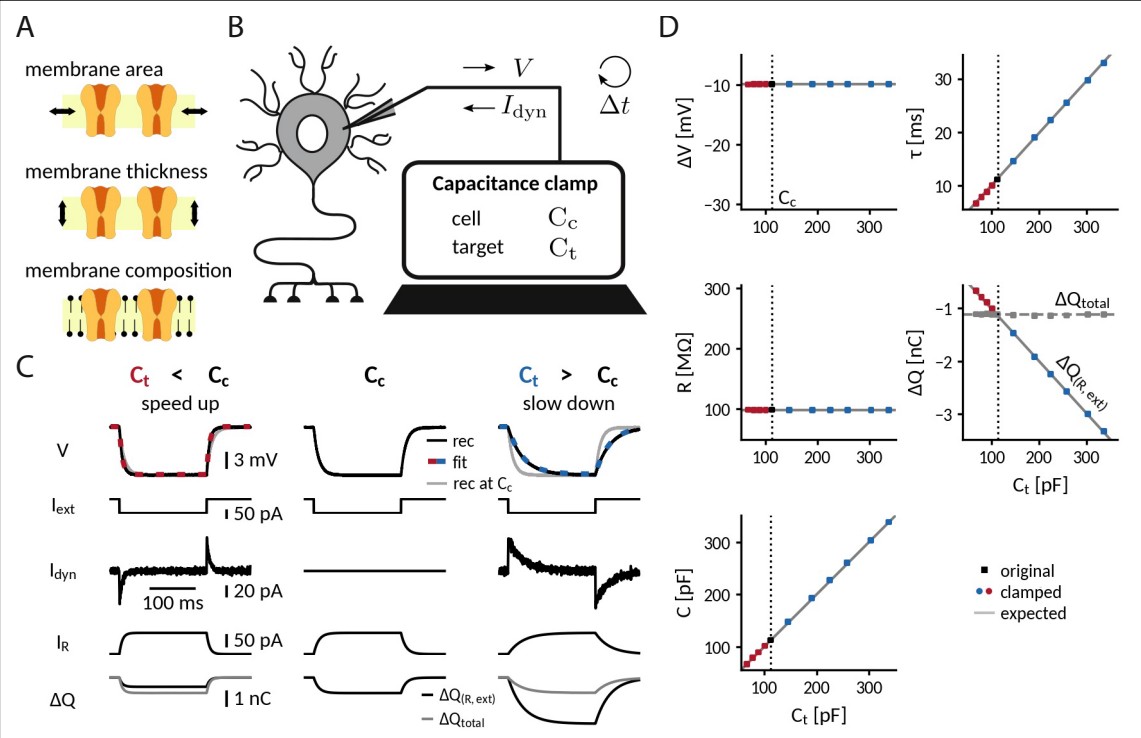

**Figure 1.** Adding or removing artificial capacitance via the CapClamp. (**A**) Physically, membrane capacitance varies with surface area, thickness and lipid composition (**B**) Virtual capacitance modification via the CapClamp is a form of dynamic clamp, a fast feedback loop between intracellular voltage sampling and computer-controlled current injection: given the measured cell capacitance $C_c$, the target capacitance $C_t$, recorded membrane potentials and the sampling interval $\Delta t$, the computer calculates clamping currents required to mimic the desired change of capacitance (see *Equation 1*). (**C**) Clamping a hardware-implemented model cell (RC circuit) at a decreased (left) or increased (right) capacitance leads to faster respectively slower charging of the 'membrane potential' $V$ to the same steady-state voltage response (top row, black: recordings, dashed red and blue: exponential fits $\Delta V \left(1 - e^{-\frac{t}{\tau}}\right)$, gray: recording at original $C_c$) in response to a step current $I_{\text{ext}}$ (2nd row) due to the clamping currents $I_{\text{dyn}}$ (3rd row). As a result, the current through the resistance $I_R = -\frac{V}{R}$ (4th row) has a different profile and the apparently deposited charge $\Delta Q(R, \text{ext}) = \int \mathrm{d}t I_R + I_{\text{ext}}$ (bottom row, black) by the 'cellular' transmembrane currents decreases, respectively, increases as expected for a capacitance change. The real deposited charge $\Delta Q_{\text{total}}$ (bottom row, gray), taking into account the clamping currents, has the same steady-state amplitude in all three cases, because the physical capacitance did not change. (**D**) Measured time constant $\tau$, voltage responses $\Delta V$, resistance $R$, deposited charge $\Delta Q$ (apparent and total) and capacitance $C$ versus target capacitances.

The online version of this article includes the following figure supplement(s) for figure 1:

**Figure supplement 1.** Impedance analysis of an RC circuit coupled to the capacitance clamp.

sufficiently high update rate $f_{\text{dyn}}$ (often ≥10 kHz), this current injected via the recording electrode makes the dynamics of the neuron appear as if the virtual channels represented by the conductance model were physically present in the membrane.

Whereas conductances gate ionic currents across the membrane, the capacitance determines how fast these currents can change the membrane potential. Every altered membrane property that results in a modified capacitance value, such as membrane area, thickness or lipid composition, affects this rate of change of the membrane potential (*Figure 1A*). To artificially mimic a modified capacitance, we therefore first asked whether a dynamic clamp protocol with its fast feedback loop between voltage sampling and current injection could adjust the 'speed' of a cell's membrane potential (*Figure 1B*). Using the current balance equation, the basic mathematical description of membrane voltage dynamics, we derived a capacitance clamp (CapClamp) scheme with a simple expression for the clamping current $I_{\text{dyn}}$ (see "Derivation of the CapClamp current" in Methods),

$$I_{\text{dyn},i} = \frac{C_c - C_t}{C_t} \left( C_c \frac{V_i - V_{i-1}}{\Delta t} - I_{\text{dyn},i-1} \right), \qquad (1)$$

which only requires the experimenter to measure the cell capacitance $C_c$ in order to set a new target capacitance $C_t$. In every sampling interval $\Delta t = f_{\text{dyn}}^{-1}$, the CapClamp uses the measured cell capacitance value $C_c$ and the voltage derivative $\frac{V_i - V_{i-1}}{\Delta t}$ to estimate the present membrane current and then increases ($C_t < C_c$) or decreases ($C_c < C_t$) the net current by insertion of a correction current in the next time bin. In this way, despite a physically unaltered capacitance, the membrane potential changes faster or, respectively, slower – as if the clamped cell actually had the different capacitance $C_t$ selected by the experimenter. In the following, we will demonstrate the CapClamp in simulated and experimental scenarios with increasing complexity ranging from a passive RC circuit up to biological neurons with a spatially extended morphology.

## Clamping capacitance in a passive cell

The simplest scenario to apply the CapClamp is a single compartment passive cell, equivalent to an RC circuit. In the absence of active conductances, the effects of a capacitance change can be precisely formulated: the capacitance $C$ sets the membrane time constant $\tau = RC$, determining how fast the membrane potential changes in response to a current. Note that, in contrast to the resistance $R$, the change in capacitance leaves the voltage amplitude of the steady-state response unaltered. To quantitatively confirm the effects of clamping capacitance and the ability of an exclusively temporal control, we measured time constant and capacitance of a clamped RC circuit in experiment and analyzed the temporal filtering properties of a modeled clamped circuit using mathematical analysis.

To experimentally characterize a clamped passive cell, we implemented the CapClamp scheme in a dynamic clamp setup (see "Dynamic clamp setup" in Methods) and recorded voltage responses to current pulses from the simplest possible model cell, that is, a hardware implemented RC circuit, while clamping it at a range of target capacitances (*Figure 1C*). As expected for an RC circuit, the charging curve of the unclamped model cell was fit well by a single exponential, whose time constant ($\tau$ = 11.1ms) and voltage amplitude ($\Delta V = -9.9$ mV) allowed us to determine the circuit's resistance $R$=99.4 MΩ and capacitance $C$=112.3 pF. This capacitance value was then used as the cell capacitance $C_c$ input for the CapClamp. Clamped at a decreased capacitance, the time constant shortened ($C_t$ = 67.4 pF: $\tau$=6.6ms) and at an increased capacitance, it lengthened ($C_t$ = 336.9 pF: $\tau$=33.0ms), but in both cases the steady state voltage amplitude remained the same. Accordingly, the measured capacitance of the clamped circuit confirmed the chosen target capacitance for the whole tested range from a 0.6- up to a 3-fold change with respect to the original capacitance (e.g. $C_t$=67.4 pF: $C$=67.5 pF; $C_t$=336.9 pF: $C$=338.1 pF), whereas the measured resistance remained constant (*Figure 1D*).

As a consequence of the correctly transformed voltage response, the leak current in the clamped RC circuit also behaved as if the capacitance had changed. When the circuit was clamped, the leak current through the resistance, $I_R = \frac{V}{R}$, exhibited a shorter ($C_t < C_c$) or longer ($C_t > C_c$) transient until reaching steady state. Further, the charge $\Delta Q(I_R, I_{\text{ext}})$ deposited on the capacitance by the apparent 'transmembrane' current, the sum of leak and external stimulus current, reduced ($C_t < C_c$) or increased ($C_t > C_c$) to the extent expected for an altered capacitance (*Figure 1C*). In contrast, the overall deposited charge $\Delta Q(I_R, I_{\text{ext}}, I_{\text{dyn}})$, including the clamping current, attained the same steady-state amplitude in the clamped and the original circuit, reflecting that the physical capacitance did not change. For the simple RC circuit considered here, the distinction between the clamping current and the intrinsic 'cellular' currents might appear artificial, because all currents use the same charge carrier. In a biological neuron, however, this distinction becomes relevant, because the clamping currents through the recording electrode might rely on other charge carriers (depending on the used intracellular solution) than the cellular currents governed by multiple ion selective channel types.

For more complex stimuli than a simple current pulse, the temporal filtering properties of a clamped membrane determine how well the CapClamp mimics the chosen capacitance change. To generally assess these filtering properties, we analytically derived the frequency-dependent impedance of a modeled clamped RC circuit using linear control theory (*Figure 1—figure supplement 1b* A, see "Impedance of a capacitance-clamped RC circuit" in Appendix 1). The derived impedance profiles confirmed the experimentally observed altered time constants. For example, an RC circuit clamped at an increased capacitance further attenuated non-zero frequencies reflecting its longer time constant. Overall, impedance amplitudes of a clamped RC and the corresponding target circuit fit well up to a tenth of the dynamic clamp frequency $f_{\text{dyn}}$, that is up to ≈2 kHz for a 20 kHz dynamic clamp system as used here (*Figure 1—figure supplement 1B and C* ). As high frequencies are heavily attenuated

by the low pass filter of a cell's membrane, these differences lead to relatively small deviations in the voltage responses. The mathematical analysis thus suggests that for a fast dynamic clamp system (> 20 kHz), the CapClamp is expected to work well for most stimuli with time scales in the physiological range.

## Simulation of the capclamp in a biophysical neuron model

In neurons with active spike-generating conductances, capacitance changes impact neuronal firing via the interplay of the altered membrane time constant and the gating kinetics of the channels involved. As gating dynamics can be in the sub-millisecond range, for example for transient sodium channels, the CapClamp is expected to require a sufficiently high dynamic clamp frequency to accurately reproduce changes of spike shape or firing rate. To understand these requirements and lay the ground for investigations of capacitance changes in biological neurons, we simulated the CapClamp in a neuron model with biophysical channel dynamics and a single-compartment morphology (see "Biophysical neuron model" in Methods). The simulation allowed us to compare the firing of the clamped neuron to the expected firing at this modified capacitance.

Specifically, we inspected the spiking responses to a depolarizing current for the original 150 pF, a decreased 90 pF and an increased 210 pF capacitance, for the latter two comparing clamped and expected dynamics (*Figure 2A*). Capacitance changes exerted a notable influence on both firing frequency and spike shape, which was mostly well-captured by the simulated CapClamp (*Table 1*). When the capacitance was decreased to 90 pF, spiking became faster and action potentials had a larger peak amplitude, a decreased duration and an increased afterhyperpolarization (AHP). When the capacitance was increased to 210 pF, the effects were opposite: spiking became slower and action potentials had a smaller peak amplitude, an increased duration and a reduced AHP. At decreased capacitances, spike amplitudes of the clamped neuron were larger than in the respective control simulation, a consequence of the limited tracking of the fast sodium current at the dynamic clamp frequency used (*Figure 2B and C*). Except for this brief overshoot, the CapClamp overall forced the membrane potential on the expected trajectory and correctly adjusted the resulting ionic currents and the gating variable dynamics of the active conductances. For example, at a reduced capacitance of 90 pF, sodium channels inactivated less during the fast rise of the AP and therefore the sodium current exhibited a second peak during AP repolarization (see the sodium inactivation variable $h$ at AP peak time in *Figure 2B*).

A subsequent comparison of simulated spiking for the whole range of tested target capacitances from 75 pF to 225 pF confirmed that the CapClamp reliably reproduced the main effects of a modified capacitance on spike shape (*Figure 2D*) and firing frequency (*Figure 2E*). Furthermore, the obtained frequency-current curves fit well with the theoretically expected reduction of excitability at higher capacitance: a decrease of gain proportional to $\frac{1}{C}$ and a constant rheobase current (see "Analytically expected effect of capacitance on the form of the f-I curve" in Methods). A crucial factor for the CapClamp, especially for a good quantitative fit of the spike shape, is the dynamic clamp frequency – observable differences at a 20 kHz sampling frequency were strongly reduced for a sampling frequency of 100 kHz (*Figure 2C and D*). In this regard, the chosen neuron model is especially demanding because its rapid gating dynamics are fit to a fast spiking interneuron (*Wang and Buzsáki, 1996*). Taken together, our simulations show that capacitance impacts neuronal spiking from firing frequency to action potential shape and that the CapClamp is well-suited to study these effects.

## Experimental demonstration of the CapClamp in rat dentate gyrus granule cells

Biological neurons differ from the simple 'cells' considered so far, that is RC circuit and single compartment neuron model, in one major aspect: they can have complex morphologies, where the membrane potential varies between different compartments and membrane capacitance is distributed across the neuronal structure. As the CapClamp in contrast operates locally through the recording electrode, the emulated capacitance change is expected to be localized to the recorded compartment instead of affecting all compartments. To demonstrate such localized capacitance changes and study their effects on neuronal spiking, we applied the CapClamp in in vitro patch-clamp recordings of rat dentate gyrus granule cells (DGGCs). Among morphologically complex cells, DGGCs appear well-suited to test the CapClamp, because their morphological structure, consisting of a central soma and

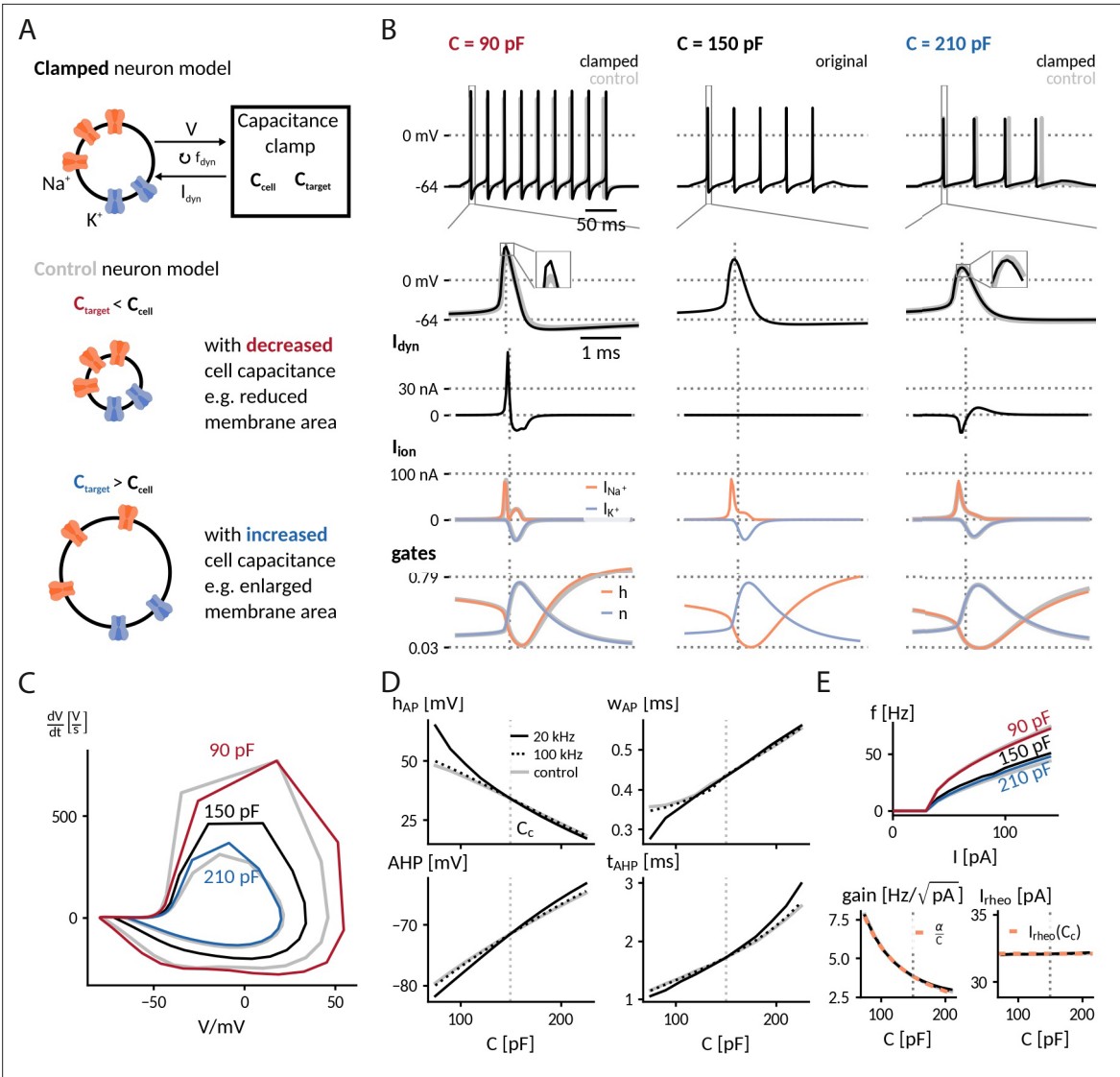

**Figure 2.** Simulation of the capacitance clamp in a conductance based neuron model. (**A**) Neurons coupled to the CapClamp are compared with control neurons with an altered capacitance (depicted as a difference in membrane area). (**B**) Spiking at 0.6-fold decreased (90 pF), original (150 pF) and 1.4-fold increased capacitance (210 pF) with from top to bottom: spike shape, dynamic clamp current, ionic currents (Na$^+$, K$^+$) and gating states (h: sodium inactivation gate, n: potassium activation gate). Clamped and original traces in black or color, control in gray. All currents are shown with the sign they appear with in the current-balance equation (**Equation 2**). (**C**) Comparison of spike shapes in the V-$\frac{dV}{dt}$-plane (black: original, red and blue: clamped, gray: control). (**D**) Comparison of spike amplitude $h_{AP}$ (top left), spike width $w_{AP}$ (top right), after hyperpolarization amplitude **AHP** (lower left) and timing $t_{AHP}$ (lower right) across different capacitances with two dynamic clamp frequencies (solid: 20 kHz, dotted: 100 kHz, gray:control). (**E**) *Top*: Comparison of f-I curves for capacitances in B. *Bottom*: Gain and rheobase current of the f-I curves across capacitances (solid: 20 kHz, dotted: 100 kHz, gray: control) compared with the theoretical predictions (orange, dashed) of a decreasing $\text{gain}(C) = \frac{\alpha}{C}$; $\alpha = \text{gain}(C_c)C_c$ and a constant rheobase (see "Analytically expected effect of capacitance on the form of the f-I curve" in Methods).

one to four primary apical dendrites as shown in *Figure 3A* (*Rihn and Claiborne, 1990*), translates to a relatively compact electrotonic structure (*Schmidt-Hieber et al., 2007*; *Wybo et al., 2019*).

## Measurement of local near-somatic capacitance
Most capacitance measurements aim to provide an accurate estimate of the *global* capacitance of a neuron (*Golowasch et al., 2009*; *White and Hooper, 2013*). To correctly infer the transmembrane and axial current, however, the CapClamp requires the *local* capacitance value of the compartment where the electrode is placed at. For the somatic DGGC recordings, we exploit that the current clamp step method – fitting charging curves via a sum of exponential terms – can also provide local capacitance

**Table 1.** Spike shape and firing frequency in a biophysical neuron model at 60 pA as well as f-I curve gain and local gain reduction for a decreased, the original and an increased capacitance, comparing simulations of an actually altered capacitance with the CapClamp. Values are shown as actual(clamped).

| C (pF) | f (Hz) | $h_{AP}$(mV) | $w_{AP}$(ms) | AHP (mV) | Gain (Hz/$\sqrt{pA}$) | Δ Gain (Hz/$\sqrt{pA}$) per 10 pF |
|---|---|---|---|---|---|---|
| decreased 90 | 34.9 (34.3) | 45.7 (55.0) | 0.30 (0.30) | −77.8 (-79.7) | 6.5 (6.5) | −0.67 (-0.67) |
| original 150 | 22.1 | 33.9 | 0.39 | −71.5 | 3.8 | −0.22 |
| increased 210 | 17.8 (18.9) | 21.4 (20.1) | 0.48 (0.48) | −66.0 (-64.7) | 2.9 (2.9) | −0.11 (-0.10) |

information (*Golowasch et al., 2009*). DGGC charging curves consisted of a slow ($\tau_0$: 15.1 ± 4.8 ms, $R_0$: 127 ± 45 MΩ) and a fast ($\tau_1$: 0.77 ± 0.24 ms, $R_1$: 35 ± 15 MΩ) component. Such a response can be understood in terms of a two compartment circuit consisting of a *near* compartment, comprising the patched soma and its surrounding, coupled to a *far*, mostly dendritic, compartment as depicted in *Figure 3A* (for details on the mapping, see "Capacitance measurements" in Methods). Importantly, the slow and fast components can be mapped to the corresponding five circuit parameters: near capacitance $C_n$ (21.0 ± 9.4 pF), near resistance $R_n$ (854 ± 394 MΩ), coupling resistance $R_a$ (53 ± 20 MΩ), far capacitance $C_f$ (106 ± 33 pF), and far resistance $R_f$ (156 ± 60 MΩ) (*Figure 3C*). Accordingly, this near-somatic capacitance $C_n$ represents the summed capacitance of the membrane area that is electrotonically close to the recording site and thus is the value that the CapClamp requires as input and should be able to modify.

## Altered near-somatic capacitance in DGGCs

To confirm the localized effect of the CapClamp, we repeated the above capacitance measurement while clamping DGGCs at values ranging from 0.6 to 3 times the original near capacitance. *Figure 3B* depicts how the charging of the membrane potential in an exemplary cell changed its shape in reaction to the clamp. Both slow and fast time constant lengthened with capacitances, whereas the associated resistances increased and decreased, respectively, such that their sum, the total input resistance (which is expected to be independent of capacitance), remained constant. These measured time constants and amplitudes matched the predicted ones for a two compartment circuit with a near capacitance at the chosen target values and all other circuit parameters at their original values. In a multicompartment simulation of a morphologically reconstructed DGGC, we could reproduce both the two compartment structure of DGGCs and the isolated modification of the near capacitance, further confirming the local control via the CapClamp.

Across 18 recorded cells, the CapClamp robustly altered DGGC charging curves and modified their charging time constants. Within the tested capacitance range, the slow time constant $\tau_0$ decreased by −0.8 (-1.0 to -0.6) ms, median and interquartile range in parentheses, and increased up to 3.0 (2.4 to 3.9) ms, whereas the fast time constant $\tau_1$ changes ranged from −0.24 (-0.29 to -0.20) ms up to 0.60 (0.36 to 0.86) ms (*Figure 3D*). To quantify how well these changes reflected an altered near capacitance, we evaluated the goodness of fit between the observed and expected time constants and resistances. In the majority of cells, R-squared values were close to 1, indicating that the CapClamp induced the expected changes ($\tau_0$: 0.87 (0.76 to 0.92), $R_0$: 0.77 (0.56 to 0.89), $\tau_1$: 0.76 (0.32 to 0.97), $R_1$: 0.85 (0.75 to 0.91)). The largest mismatches occurred for the fast time constant, especially at high capacitances, where the measured time constant was often shorter than predicted (*Figure 3D*). A small bias toward a shorter fast component is to be expected and also present in the multicompartment simulation, because this time constant was only about ten times longer than the sampling interval of 50 μs limiting its slowing-down by the CapClamp currents. Larger deviations of $\tau_1$ however could not be reproduced in numerical simulations and likely result from other error sources, such as the difficulty of fitting this small and short time constant in the presence of noise or imprecise estimates of the original near capacitance (see "Online measurement of capacitance" in Methods). Overall, in terms of circuit parameters, the capacitance measurements confirmed the targeted near capacitance change for 12 out of 18 cells within an average error of 10% (*Figure 3E*). In summary, the CapClamp achieved an isolated change of the near-somatic capacitance in DGGCs and thereby allows to control the time constants of their passive voltage dynamics.

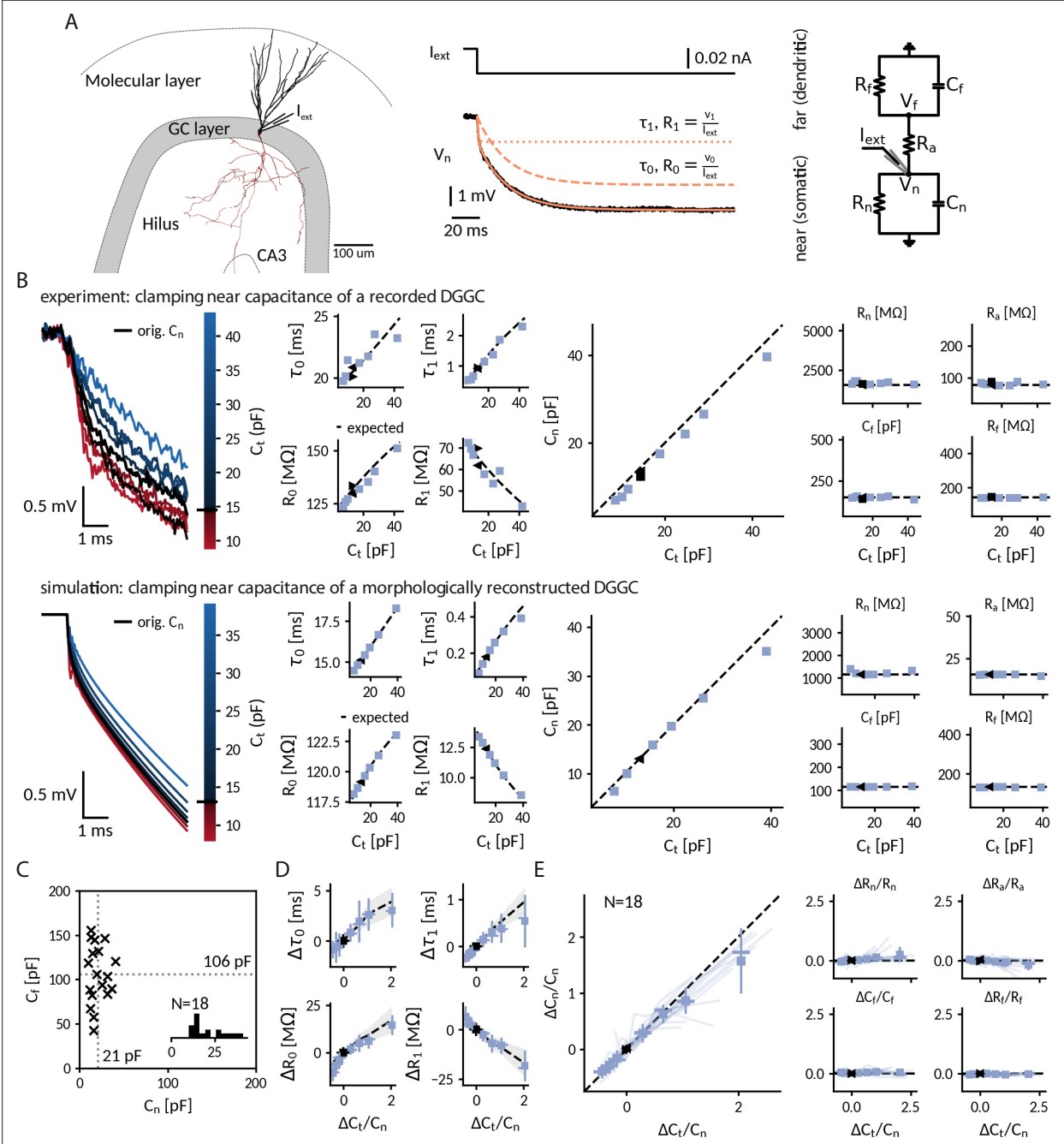

**Figure 3.** Clamping capacitance in rat dentate gyrus granule cells (DGGCs). (**A**) Morphology of a DGGC (left) and response to a hyperpolarizing current injected at the soma, fit via a sum of exponential terms with a slow $\tau_0$, $v_0$ and a fast component $\tau_1$, $v_1$ (middle), which can be mapped to two resistively coupled RC-circuits (right) with a near (somatic) compartment $C_n$ and $R_n$, resistive coupling $R_a$ and a far (dendritic) compartment $C_f$ and $R_f$. (**B**) *Left*: Voltage responses of a recorded (top) and a simulated morphologically-reconstructed (bottom) DGGC to a current pulse (exp: −27 pA, sim: −50 pA) clamped at 0.6- to 3-fold the cell's near capacitance (black: original near capacitance, color: target near capacitances). *Middle*: Slow and fast components versus target capacitance. *Right*: Circuit parameters versus target capacitance. ◀, ▶: before and after clamping, blue square: clamped, dashed line: expected values. (**C**) Measured near $C_n$ and far $C_f$ capacitances for 18 DGGCs (gray dotted: mean). Inset: histogram of near capacitances. (**D**) Changes of slow and fast components in all recorded cells versus relative targeted change of near capacitance (squares: mean, horizontal line: median, vertical line: std, shaded area: std of expected changes). (**E**) Relative changes of circuit parameters versus relative targeted change of near capacitance. Legend same as in *D* and individual cells shown with transparent blue lines.

# Near-somatic capacitance governs action potential shape and firing frequency in dentate gyrus granule cells

In neurons such as the recorded DGGCs, where the axon directly emerges from the soma, the ability to clamp the near-somatic capacitance provides control over the major capacitive load for the action potential generating site in the axon initial segment. Consequently, the CapClamp, although acting locally, is expected to impact action potential (AP) dynamics and excitability of a morphologically complex DGGC as demonstrated earlier for the simplified single-compartment neuron model (*Figure 2*). To illustrate how the CapClamp can be applied to characterize neuronal firing, we compared spiking responses and f-I curves across near capacitances ranging from 0.6 to 3 times the original value, corresponding to a range from 10 pF to 60 pF for the near and from 110 pF to 160 pF for the total (near and far) capacitance.

Clamping the near-somatic capacitance in DGGCs, we observed pronounced changes in the spiking response to depolarizing current step, clearly visible in the raw voltage traces (*Figure 4A*). The most apparent change was an altered AP shape (*Figure 4B*) – a continuous reduction of AP peak amplitude (from 60 ± 10 mV at 0.6 $C_n$ to 22 ± 17 mV at 3 $C_n$ for 9 DGGCs) and a simultaneous broadening of AP width (from 0.78 ± 0.15 ms at 0.6 $C_n$ to 1.33 ± 0.48 ms at 3 $C_n$) with increasing capacitance (*Figure 4C and D*). In addition, fast afterhyperpolarization (fAHP) was diminished and disappeared in the majority of cells after increasing capacitance (fAHP in 8/9 cells at 0.6 $C_n$ and 2/9 at 3 $C_n$). Importantly, the observed disappearance of fAHP cannot be explained by increased capacitive filtering alone, as an increased capacitance would reduce the fAHP amplitude, but not abolish it. Thus, our data suggests that the somatic capacitive load in DGGCs is able to influence the AP generating currents.

To illustrate the interplay of capacitance and the AP generating currents, we compared the observed spikes with hypothetical ones obtained by assuming unaltered currents with respect to those at the original capacitance (see "Protocol 2: Analysis of f-I curves and spike shapes" in Methods). Recorded and hypothetical spike shapes exhibited marked differences (*Figure 4B*). At 0.6-fold decreased capacitances, for example, the recorded AP amplitude was significantly smaller than the hypothetical one (rec.: 60 ± 10 mV, hyp.: 94 ± 19 mV, one-sided Wilcoxon signed-rank $Z$=0, p<0.001), presumably reflecting a reduction of the driving force for the sodium current when the AP peak approaches the reversal potential of sodium. Furthermore, at threefold increased capacitance, as noted above the recorded spikes exhibited no fAHP in most cells while the hypothetical ones still did (fAHP rec: 2/9, hyp: 8/9) – potentially a result of a reduced activation of potassium channels due to lower AP amplitudes and/or earlier closing during the slowed AP repolarization. In contrast to driving force and gating dynamics, the channel kinetics, for example their activation curves, cannot be altered by capacitance. Correspondingly, the spike threshold, which reflects the voltage where sodium channels start to massively open, was not significantly correlated with near capacitance (Pearson correlation $r$ = 0.10, p = 0.42). Taken together, our analysis indicates that an altered somatic capacitance affects both sodium and potassium currents underlying APs in DGGCs.

Near-somatic capacitance also impacted DGGC excitability. With increasing capacitance, DGGCs became less excitable and firing frequencies significantly decreased (*Figure 4D and F*). From 0.6- to 3-fold of the original near capacitance, the decrease was modest for low firing rates close to threshold (from 9.7 ±3.2 Hz to 7.8 ±3.9 Hz, Wilcoxon signed-rank $Z$=45, p=0.002) and became more pronounced for high firing rates at the largest injected currents (from 23.3 ±6.4 Hz to 18,6 ±4.8 Hz, $Z$=45, p=0.002). In terms of the firing rate-current (f-I) curves, the gain of the DGGCs significantly decreased with capacitance (from 1.82 ± 0.40 Hz/$\sqrt{\mathrm{pA}}$ at 0.6 $C_n$ to 1.48 ± 0.34 Hz/$\sqrt{\mathrm{pA}}$ at 3 $C_n$, $Z$=45, p=0.002), whereas the rheobase current remained relatively constant (from 185 ±82 pA at 0.6 $C_n$ to 184 ±77 pA at 3 $C_n$, two-sided, $Z$=12, p=0.25). Across cells, the gain reduction obtained by linear regression was –0.10 (-0.13 to -0.06) Hz/$\sqrt{\mathrm{pA}}$ per 10 pF near capacitance (median and interquartile range, significant slope in 8/9 cells, p < 0.1). Compared to the simulated neuron with a gain reduction of –0.22 Hz/$\sqrt{\mathrm{pA}}$ per 10 pF over the same capacitance range as in the DGGC experiments (see *Figure 2D*), DGGCs thus exhibit a weaker gain dependence on near capacitance. A biological factor for this reduced effect in the DGGCs is their overall smaller gain – reflecting the different set of ionic conductances compared to the Wang-Buzsáki model designed to mimic a fast-spiking cortical interneuron. Assuming a scaling of gain with 1/C (as predicted theoretically for neurons with a continuous f-I curve, see "Analytically expected effect of capacitance on the form of the f-I curve" in Methods), the gain reduction is expected to be -gain($C_c$)/$C_c$, which is approximately twice as high for the model

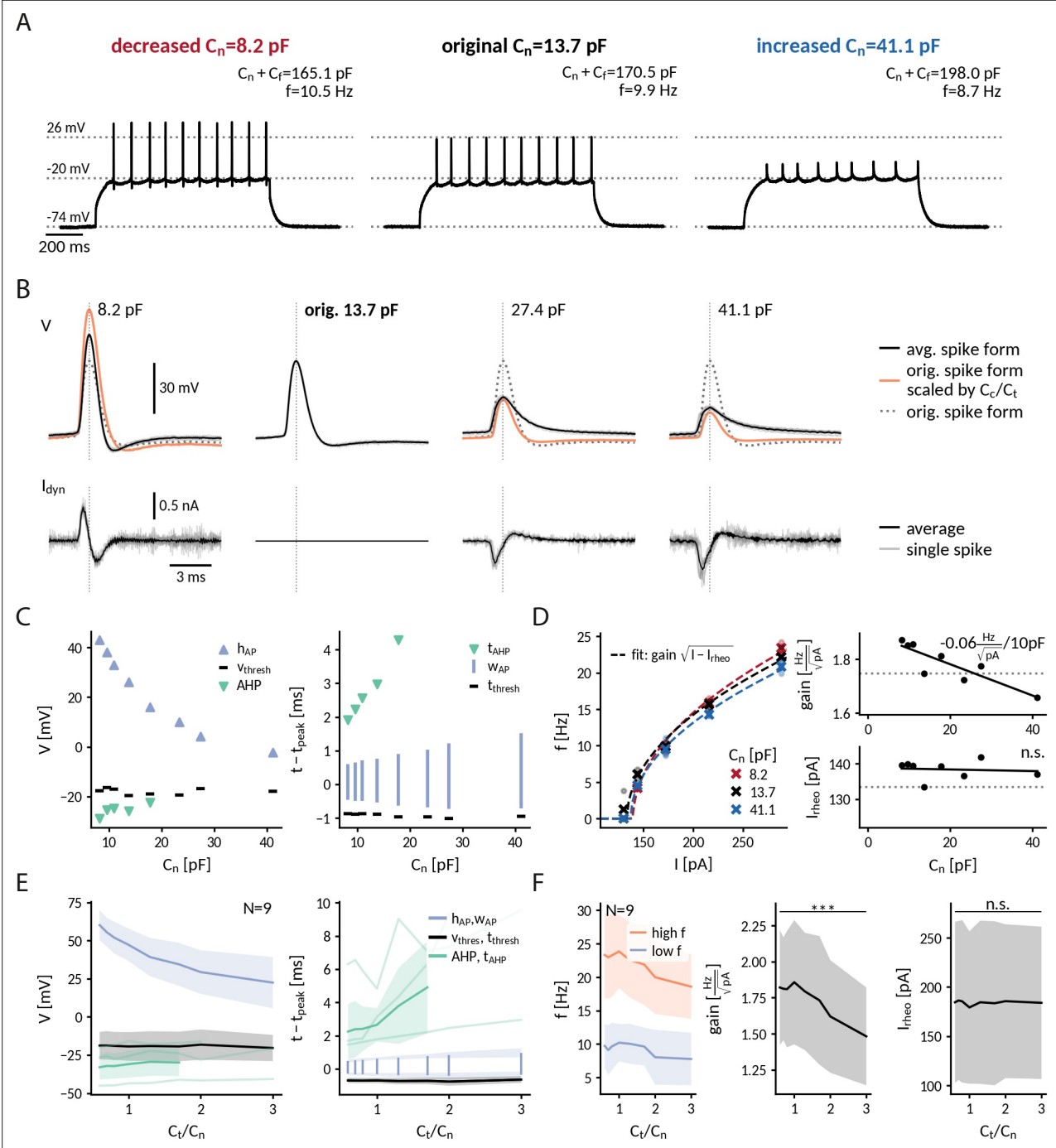

**Figure 4.** Repetitive spiking and action potential shapes in DGGCs clamped at different capacitances. (**A**) Spiking at decreased 0.6-fold (left), original (middle) and increased 3-fold (right) near capacitance $C_n$. (**B**) Spike shapes (top) and capacitance clamp currents (bottom) for increasing capacitances from 0.6 to 3-fold of the original near capacitance (black: mean, light gray: single spikes, orange: expected spike shape for unaltered intrinsic currents as described in protocol-2-analysis-of-f-i-curves-and-spike-shapesMethods, dotted: spike shape at original capacitance). (**C**) Comparison of spike shape (left) and temporal structure (right) across tested near capacitances. (**D**) Measured f-I curve at 0.6-, 1- and 3-fold near capacitance with fit $f = \text{gain}\sqrt{I - I_{\text{rheo}}}$ (dashed lines). Extracted gain and rheobase for all tested near capacitances (dotted line: values at original capacitance 13.7 pF, solid line: linear regression with slope value reported if significantly different from zero p<0.1). (**E**) Effect of near capacitance changes on spike shape (left) and temporal structure (right) for all recorded DGGCs (solid: mean, shaded: std). To compare different cells, the capacitance is shown relative to the original near capacitance and spikes were compared at 1.2-fold of the cell's rheobase. (**F**) Effect of near capacitance changes on firing frequency, low firing (blue) at 1.2 fold rheobase and high firing (red) at 2.0-fold rheobase (left), gain (middle) and rheobase (right) for all recorded DGGCs (solid: mean, shaded: std).

$(\approx -\frac{3.8\frac{\text{Hz}}{\sqrt{\text{pA}}}}{150\text{pF}} = -0.25\ \text{Hz}/\sqrt{\text{pA}}$ per 10 pF) compared to the average DGGC $(\approx -\frac{1.8\frac{\text{Hz}}{\sqrt{\text{pA}}}}{127\text{pF}} = -0.14\ \text{Hz}/\sqrt{\text{pA}}$ per 10 pF). A further technical factor for a weaker effect in the DGGCs is the local nature of the capacitance modification. Depending on the particular location and geometry of the axon initial segment, the influence of the clamped somatic compartment on AIS excitability can differ (*Goethals and Brette, 2020*). The altered excitability in the majority of DGGCs, however, demonstrates that clamping their near capacitance was sufficient to affect the capacitive load of their AIS. We conclude that a somatic capacitance clamp, altering perisomatic capacitance alone, is able to modify the input-output relationship of a real neuron.

## Applications of the CapClamp

The CapClamp lends itself to either test hypotheses on the impact of capacitance or to exploit the control over the membrane time constant in order to to alter neuronal dynamics in informative ways. In the following, we briefly illustrate applications of the CapClamp from these two fields, applying the

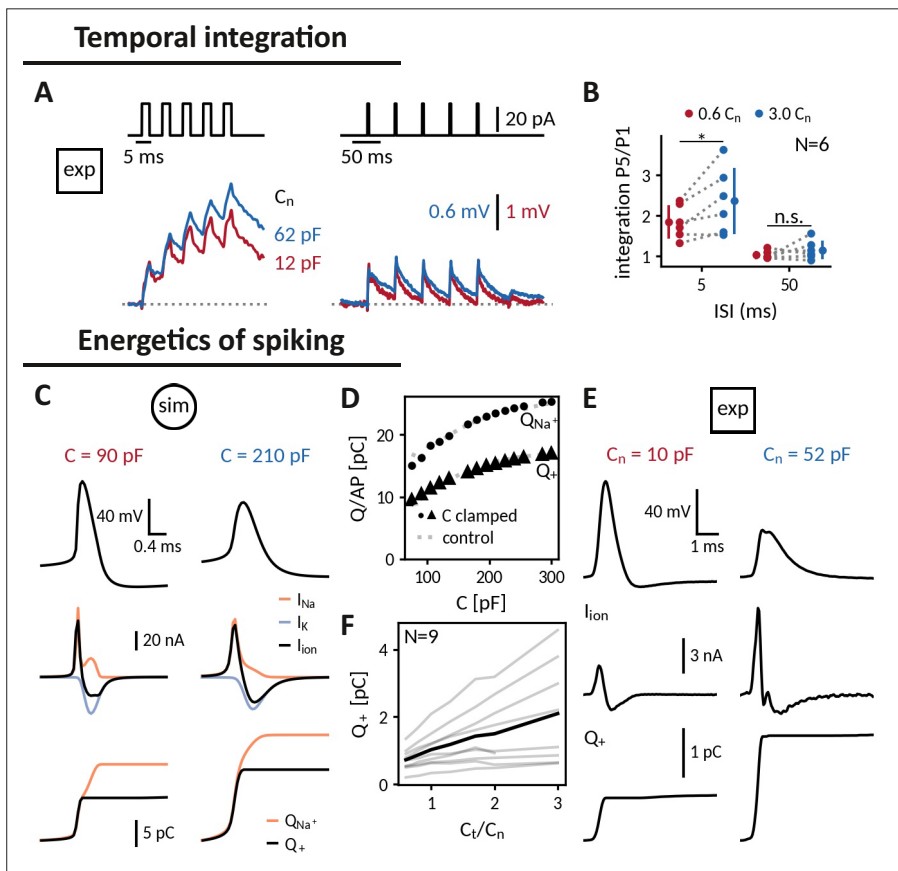

**Figure 5.** Applying the capacitance clamp to study neuronal signaling and physiology. Temporal integration: (**A**) Brief current pulses of 3ms length with interstimulus intervals of 5ms and 50ms (top) and voltage responses of an exemplary DGGC at a decreased (12 pF) and an increased (62 pF) near capacitance (voltage scale adapted to first response height). (**B**) Ratio of fifth and first response as a measure of temporal integration for a 0.6-fold decreased capacitance in comparison to a threefold increased one at 5ms and 50ms ISI. Energetics of spiking: (**C**) Spike shape (top), sodium, potassium and total ionic current (middle, shown with the sign they appear with in the current-balance equation, see *Equation 2*) and deposited sodium $Q_{Na_+}$ as well as depolarizing $Q_+$ charge (bottom) in the Wang-Buzsáki neuron model for a 90 pF and a 210 pF capacitance. (**D**) Sodium $Q_{Na_+}$ and depolarizing $Q_+$ charge per action potential versus capacitance (dot and triangle: clamped from an original capacitance of 150 pF, gray: control). (**E**) Spike shape and depolarizing charge for a dentate gyrus granule cell clamped at decreased 10 pF and increased 52 pF near capacitance. (**F**) Deposited depolarizing charge versus relative change of near capacitance in recorded DGGCs (black: mean, gray: individual cells).

technique to experimentally explore effects of capacitance on temporal integration and energetic costs of spiking.

## Temporal integration

A basic processing step in neuronal computation is temporal integration, the summation of time-separated synaptic inputs (*Krueppel et al., 2011*; *Athilingam et al., 2017*). An upper limit for temporal integration, at least in the absence of dedicated active channels, is set by the membrane time constant $\tau = RC$, which is directly proportional to the cell's capacitance. Hence, increasing the capacitance of a cell should make it a better integrator: if two brief inputs arrive separated by less than the membrane time constant, the cell will summate the responses so that the membrane potential after the second is higher than after the first one. Indeed, when we compared the responses of DGGCs clamped at decreased and increased near-somatic capacitances to current pulse trains, increasing the capacitance allowed the cell to better "sum" 3ms pulses at an inter stimulus interval (ISI) of 5 ms as apparent by the stair-like voltage response with a higher ratio of last to first pulse response. (*Figure 5A and B*). At an ISI of 50ms, in contrast, neither capacitance was sufficient for temporal integration. The biological relevance of tailoring capacitance to temporal processing can, for example, be observed in auditory cells of the barn-owl, which have no dendrites to reduce capacitive and resistive load and hence shorten their time constant such that they can perform sub-millisecond coincidence detection (*Ashida et al., 2007*).

## Energy consumption during spiking

Action potentials are energetically expensive, because the contributing sodium and potassium ions need to be pumped back using ATP (*Laughlin et al., 1998*; *Hasenstaub et al., 2010*). The minimal amount of ionic charge required for an action potential is dictated by the capacitance as $Q = C\Delta V_{AP}$, suggesting that a smaller capacitance is energetically favorable. In order to gauge how capacitance affects charge accumulation and energy consumption, we reexamined spike shapes for a fixed current input at different capacitances both in the simulated neuron and in the recorded DGGCs (*Figure 5C and E*). We found that despite a reduced amplitude at larger capacitances, these smaller spikes still required more depolarizing charge $Q_+ = C\Delta V_{AP}$ (*Figure 5D and F*). In the model, we tested whether this depolarizing charge provided a reliable indication of the sodium charge $Q_{Na^+}$, which finally determines pump activity and energy consumption (*Figure 5D*). Due to the overlap of sodium and potassium currents, the sodium charge exceeded the net depolarizing charge, but as this overlap remained roughly constant, both charge measures increased by the same amount with capacitance. Taken together, in the tested model and the recorded DGGCs, energy consumption per action potential appears to be reduced at smaller capacitances. In line with this observation, it has been reported that perineuronal nets could decrease membrane capacitance of fast-spiking interneurons, thereby facilitating high-frequency firing, while keeping energetic costs at bay (*Tewari et al., 2018*).

## Discussion

The dynamic clamp is a valuable tool in intracellular recordings to examine the diverse roles of ionic conductances in excitable cells (*Sharp et al., 1993*; *Prinz et al., 2004*; *Wilders, 2006*; *Economo et al., 2010*). In this study, we introduced the capacitance clamp (CapClamp), an application of the dynamic clamp that allows electrophysiologists to mimic a modified membrane capacitance in a biological neuron. Via simulations of a biophysical neuron model, we confirmed that the CapClamp correctly captures how capacitance affects spike shapes and firing frequency. In recordings of rat dentate gyrus granule cells, we further verified that the CapClamp could accurately control the capacitance of the recorded somatic compartments. Moreover, we clamped this near-somatic capacitance of DGGCs during spiking and found that, as predicted by our simulations, capacitance can modify the fI curve and alter the course of the spike generating currents. CapClamp can serve as a new probe to neuronal signaling and physiology. In the following, we highlight requirements for the CapClamp and discuss how this experimental control over capacitance can benefit the study of cellular electrical behavior.

## Precise, flexible and local control over capacitance in all excitable cells

To our knowledge, the CapClamp is the first tool to experimentally study capacitance changes in a precise and flexible manner. The CapClamp owes its precision and flexibility to the virtual nature of the altered capacitance. In contrast, methods to physically modify the capacitance are affected by various undesired side effects. Dendritic pinching, decoupling dendrites from the soma, for instance greatly reduces membrane area and thereby capacitance, but also removes all dendritic conductances (*Bekkers and Häusser, 2007*). Capacitance alterations have also been reported after application of mefloquine, a drug binding to membrane phospholipids, but it also blocks gap junctions (*Szoboszlay et al., 2016*). A notable exception is the recent demonstration of engineered polymer synthesis in neuronal cell membranes, which alters their capacitance, but not their input resistance (*Liu et al., 2020*). In comparison, however, the CapClamp provides more accurate and dynamic control by allowing experimenters to test multiple selected capacitance values in a single cell (*Figures 3 and 4*), while being significantly simpler to implement.

The CapClamp can be applied in every excitable cell. Here, we focused on neurons, but the proposed clamping currents can also be used to study capacitance changes in other cells, including for example heart cells (*Wilders, 2006*; *de Oliveira et al., 2015*). In particular, no prior knowledge about the ionic or external currents in the clamped cell is required, so that the capacitance can be clamped during any experimental protocol (step current, ramp current, etc.) or during synaptic input. Furthermore, capacitance can be clamped in both electrotonically compact cells like oocytes (*Ori et al., 2020*) and non-compact cells like most neurons (*Wybo et al., 2019*), although in the latter case the CapClamp is limited locally to the capacitance of the recorded compartment (*Figure 3*). Consequently, the effects of clamping capacitance depend on the cell's morphology and the recording site. The soma, for example, represents the major capacitive load for spike generation in vertebrate neurons, where the axon predominantly emerges close to the soma (*Figure 4*), but it is expected to exert less influence in neurons, where the axon comes out of the dendritic tree, a common feature of invertebrate neurons (*Hesse and Schreiber, 2015*), but also seen in mammalian neurons (*Martina et al., 2000*; *Thome et al., 2014*).

The major prerequisite to apply the CapClamp is a reliable capacitance measurement of the clamped compartment, which can be challenging, especially for electrotonically complex cells (*Golowasch et al., 2009*; *White and Hooper, 2013*). An imprecise capacitance estimate leads to erroneous clamping currents, which increase high-frequency noise for small errors and might even induce instabilities for larger errors. The measurement method presented for the recorded DGGCs, that is mapping the charging response to a two compartment circuit, could in principle be extended to cells with a larger number of compartments e.g. pyramidal cells (*Edwards and Mulloney, 1984*; *Wybo et al., 2021*). Yet, accurate multi-exponential fitting is demanding and the assumption of uniform membrane properties underlying the mapping is a simplification, shown to be violated in some cells, such as GABAergic interneurons (*Nörenberg et al., 2010*). As an alternative, measurement protocols could be exploited that inherently yield local capacitance estimates, including fast voltage ramps (*Golowasch et al., 2009*) or sampling of voltage responses to fast fluctuating currents (*Badel et al., 2008*). Reliable capacitance measurements can further be used to compare measured and target capacitance of the clamped cell, which can serve as a first simple test to ensure the quality of the CapClamp.

## A CapClamp on every rig

As a novel application of the established dynamic clamp technique, the CapClamp is an accessible and low-cost extension of a standard electrophysiology stack (*Prinz et al., 2004*; *Economo et al., 2010*). For an existing dynamic clamp setup, the sole requirement is to implement the calculation of the clamping currents (see *Equation 5*). Otherwise, multiple open source frameworks exist that only require a dedicated computer with a data acquisition card to enable the dynamic clamp in a conventional electrophysiology setup (*Dorval et al., 2001*; *Benda et al., 2007*; *Kemenes et al., 2011*; *Linaro et al., 2015*; *Patel et al., 2017*; *Desai et al., 2017*; *Amaducci et al., 2019*). To facilitate the usage of the technique, we provide code for the CapClamp scheme in the RELACS and RTXI frameworks (see "Data and software availability" in Appendix 1).

In CapClamp recordings, as in all dynamic clamp applications, a high sampling frequency and accurate voltage monitoring are key (*Bettencourt et al., 2008*). Whether a sampling frequency is

sufficiently high can be tested by assuring that the observed voltage dynamics for example the spike amplitudes are invariant when the sampling frequency is decreased from the maximal possible value (*Robinson, 1994*). For the simulated fast-spiking interneuron, we found a satisfactory clamp at a frequency of 20 kHz, which we expect to also be sufficient for most excitatory neurons, because they tend to have slower voltage dynamics (*Hasenstaub et al., 2010*). In our single electrode recordings, we focused on careful electrode compensation to avoid electrode artifacts in the recorded voltages which would lead to incorrectly estimated membrane currents and eventually instabilities. To improve voltage monitoring, future applications could either apply active electrode compensation (*Brette et al., 2008*; *Bal and Destexhe, 2009*) or resort to two electrode recordings, where current injection and voltage recordings are separated.

## Modifying capacitance as a probe for cellular electrical dynamics

Via the CapClamp, experimenters can ask a question that was previously accessible only in theoretical work: What if capacitance was different? In contrast to the theoretical approach, the answers to this question do not have to rely on models of channel dynamics or other membrane properties, because the latter are provided by the biological cell itself (*Sharp et al., 1993*). Modifying capacitance with the CapClamp can serve either to investigate changes in this biophysical parameter or, more broadly, to alter the membrane time constant of a cell as a way to characterize its electrical dynamics.

### Understanding the role of capacitance

The virtual capacitance changes induced by the CapClamp could serve to address two crucial questions about actual membrane biophysics: why capacitance appears to be biologically mostly constant (*Gentet et al., 2000*) and how exceptions to this rule can facilitate or deter neuronal function (*Amzica and Neckelmann, 1999*; *Hartline and Colman, 2007*; *Eyal et al., 2016*; *Tewari et al., 2018*). Capacitance is for example rarely tested for optimality - a common question in ion channel kinetics, which appear optimized for function and energy expenditure (*Hasenstaub et al., 2010*; *Sengupta et al., 2010*). Regarding energy consumption, our CapClamp experiments in DGGCs indicate that action potentials become energetically cheaper at lower capacitances (*Figure 5E and F*). Interestingly, reports of exceptional capacitance values mostly find reductions e.g. for myelinated axons ($C_m \approx 0.05$ uF/cm$^2$ for a 10-fold wrapped myelin sheath, see *Castelfranco and Hartline, 2015*) or human pyramidal cells ($C_m \approx 0.5$ uF/cm$^2$, see *Eyal et al., 2016*) suggesting that indeed the metabolic cost of AP generation could have been a contributing factor to capacitance adaptations. In addition, the recent hypothesis that perineuronal nets can reduce capacitance of interneurons in a similar way as myelination of axons suggests that capacitance adaptation could be more widespread in the brain than often assumed (*Tewari et al., 2018*). Moreover, understanding the role of capacitance can contribute to an improved understanding of infrared (*Shapiro et al., 2017*; *Carvalho-de-Souza et al., 2018*) and ultrasonic (*Krasovitski et al., 2011*; *Plaksin et al., 2014*) stimulation of neural activity, whose effects are assumed to rely on rapid alteration of the capacitance.

Another application of the CapClamp might be to investigate changes of excitability associated with neuronal growth. During development, cell size can increase considerably, necessarily accompanied by a larger membrane capacitance (*McComb et al., 2003*). To maintain neural function, neurons need to compensate for this altered capacitance via a corresponding regulation of ionic conductances – a homeostatic process that is hypothesized to involve activity-dependent channel expression (*Gorur-Shandilya et al., 2020*). To disentangle the contributions of capacitance in this concurrent alteration with ionic conductances, the CapClamp could be combined with the 'classic' dynamic clamp, for example parallel changes of both capacitance and a leak conductance.

### Altering the membrane time constant

A key contribution of the CapClamp is the isolated experimental control of the membrane time constant via changes in capacitance while leaving the ion channel conductances unaffected. In neuron models, monitoring response properties when changing the membrane time constant has been used to characterize a cell's dynamical repertoire (*Kirst et al., 2015*; *Hesse et al., 2017*; *Franci et al., 2018*). As an experimental analogue, the CapClamp introduces this option for the characterization of biological neurons.

To optimally support neural processing, nerve cells exhibit qualitatively different response properties, which in some cases can be flexibly adapted to context. For example, neurons with class 2 excitability (marked by a jump of the f-I curve to non-zero frequencies when exceeding threshold) can be switched to class 1 excitability (marked by a smooth transition with arbitrarily low frequencies) via neuromodulation (*Stiefel et al., 2008*; *Stiefel et al., 2009*), transforming them from resonators to integrators. These qualitative differences in response and processing properties can be characterized by bifurcation analysis (*Izhikevich, 2006*; *Prescott et al., 2008a*; *Kirst et al., 2015*; *Hesse et al., 2017*). Capacitance as a canonical parameter can induce transitions between excitability classes and the underlying bifurcation types, including the switch of neuronal dynamics from class 1 excitability (with regularly spiking neurons) to dynamics that include bistable firing with stochastic switches between spiking and rest (*Hesse et al., 2017*). Because computational properties can be expected to change with such qualitative switches in dynamics, it may be of interest to determine how close the dynamics of a given cell is to a transition. An estimate of this proximity to switches that can be obtained via the CapClamp by monitoring firing properties and qualitative changes thereof as a function of membrane capacitance. Dynamics in the vicinity of capacitance-induced switches are likely to be also susceptible to switches induced by other parameters with similar temporal effects, such changes in temperature (*Hesse et al., 2017*) or ionic concentrations (*Contreras et al., 2020*). As such switches can involve regimes of exceptionally fast dynamics, for such measurements extra care should be given to ensure that the temporal resolution of the dynamic clamp is sufficiently high.

In addition to such qualitative changes of dynamics, the broad impact of the time constant (and therefore the capacitance) on firing frequency and spike shape could be applied for more quantitative studies of neuronal activity. On the one hand, observations of neural activity at different capacitances could for example be used to further constrain and improve fitting of conductance-based neuron models (*Podlaski et al., 2017*; *Gouwens et al., 2018*; *Franci et al., 2018*). On the other hand, it could serve to examine activity-dependent physiological processes such as ion concentration dynamics (*Contreras et al., 2020*) or calcium controlled channel homeostasis (*O'Leary et al., 2014*; *Temporal et al., 2014*; *Santin and Schulz, 2019*).

## Conclusion

Taken together, the presented CapClamp enables an accurate and flexible control over capacitance in biological neurons, a basic determinant of cellular excitability, that so far has been inaccessible in experiment. We expect that the CapClamp will, therefore, broaden and enrich the electrophysiological study of neurons and other excitable cells. With expanding techniques to sense and manipulate neural activity, the combination of modeling and targeted closed-loop feedback that underlies the CapClamp (and more generally the dynamic clamp *Chamorro et al., 2012*) will further unlock experimental control over other previously inaccessible aspects of single neuron (*Ullah and Schiff, 2009*; *Rivera et al., 2015*; *Harrigan et al., 2018*) and network dynamics (*Newman et al., 2015*; *Hocker and Park, 2019*).

# Materials and methods

**Key resources table**

| Reagent type (species) or resource | Designation | Source or reference | Identifiers | Additional information |
|---|---|---|---|---|
| Strain, strain background (*Rattus norvegicus*, male and female) | Wistar Rat (wild type) | Wistar Institute of Philadelphia, Pennsylvania | | |
| Peptide, recombinant protein | Avidin conjugated AlexaFluor-647 | Thermo Fisher Scientific | RRID:AB_2336066 | |
| Software, algorithm | Fiji distribution of ImageJ software | imagej.net | RRID:SCR_003070 | |
| Software, algorithm | Neutube | neutracing.com | https://doi.org/10.1523/ENEURO.0049-14.2014 | |
| Software, algorithm | RELACS | relacs. sourceforge.net | RRID:SCR_017280 | |

*Continued on next page*

*Continued*

| Reagent type (species) or resource | Designation | Source or reference | Identifiers | Additional information |
|---|---|---|---|---|
| Software, algorithm | RELACS CapClamp | This paper | https://doi.org/10.5281/zenodo.6322768 | Capacitance clamp code for RELACS, see "Data and software availability" in Appendix 1 |
| Software, algorithm | RTXI | rtxi.org | https://doi.org/10.1371/journal.pcbi.1005430 | |
| Software, algorithm | RTXI CapClamp | This paper | https://doi.org/10.5281/zenodo.5553946 | Capacitance clamp code for RTXI, see "Data and software availability" in Appendix 1 |
| Software, algorithm | Brian 2 | brian-team/ brian2 | https://doi.org/10.7554/eLife.47314 | |

## Derivation of the capclamp current

In order to derive a dynamic clamp feedback scheme for the CapClamp, we compare the actual membrane potential dynamics at the original capacitance $C_c$ with the target dynamics at the chosen capacitance $C_t$. The actual dynamics of the cell, which for the moment is assumed to be isopotential, is given by the current-balance equation of a single compartment

$$\frac{dV}{dt} = \frac{I(V,t) + I_{\text{dyn}}(t)}{C_c},$$ (2)

with capacitance $C_c$, membrane currents $I(V,t)$ (comprising all ionic and synaptic currents, as well as external stimuli) and the dynamic clamp current $I_{\text{dyn}}(t)$. Note that ionic and synaptic contributions to the membrane currents $I(V,t)$ are voltage-dependent, both with respect to driving force and gating dynamics, so that a voltage trajectory governed by a different capacitance also leads to a modified shape of the membrane currents. In the target dynamics, the dynamic clamp current is absent and the capacitance is modified to the desired value

$$\frac{dV}{dt} = \frac{I(V,t)}{C_t}.$$ (3)

Both membrane potential trajectories would coincide, if we chose a dynamic clamp current such that the right-hand sides of actual (*Equation 2*) and target dynamics (*Equation 3*) become identical,

$$I_{\text{dyn}}(t) = \frac{C_c - C_t}{C_t} I(V,t).$$

Generally, an exact model for the membrane currents $I(V,t)$ will not be available, as it would require knowledge about all active conductances and incoming synaptic inputs. Instead, the membrane current can be estimated from the stream of incoming voltage data using the discrete version of *Equation 2*

$$I(V_{i-1}, t_{i-1}) \approx C_c \frac{V_i - V_{i-1}}{\Delta t} - I_{dyn, i-1}$$ (4)

where $\Delta t$ is the sampling interval. A prerequisite is the measurement of the cell capacitance $C_c$. Furthermore, for the estimation to be accurate, the samplin ginterval needs to be shorter than the fastest time scales of changes in the membrane currents for example sodium gating time constants. With this estimated membrane current, the complete expression for the CapClampcurrent reads

$$I_{\text{dyn},i} = \frac{C_c - C_t}{C_t} \left( C_c \frac{V_i - V_{i-1}}{\Delta t} - I_{\text{dyn},i-1} \right).$$ (5)

The above derivation assumes that the cell is isopotential. In the case of an electrotonically non-compact cell, the steps are identical, but the cell capacitance $C_c$ has to be replaced by the capacitance of the compartment where the recording electrode is located. Consequently, in a non-isopotential neuron, the mimicked capacitance modification is restricted to the compartment at the tip of the

recording electrode - a constraint known as the space clamp that is shared by all clamping techniques (**Prinz et al., 2004**; **Bar-Yehuda and Korngreen, 2008**).

The indexing above assumes a voltage sampling $V_i = V(i\Delta t)$ and a quasi-immediate current injection $I_{dyn,i} = I_{dyn}(i\Delta t)$. However, sampling can take a non-negligible amount of time, so that depending on the sampling system the currently available voltage actually represents the voltage from the previous cycle $V_i = V((i-1)\Delta t)$. In this case, for a correct estimation of the membrane currents, the dynamic clamp current index has to be shifted correspondingly to $I_{dyn,i} = \frac{C_c - C_t}{C_t}\left(C_c \frac{V_i - V_{i-1}}{\Delta t} - I_{dyn,i-2}\right)$.

## Capacitance measurements

To apply the CapClamp, a prerequisite is to measure the capacitance of the recorded local compartment. Here, we use the current clamp protocol, which estimates the capacitance from the voltage response to a current step with amplitude $I_{ext}$,

$$V(t) = \sum_i v_i \left(1 - e^{-\frac{t}{\tau_i}}\right) = I_{ext}\sum_i R_i \left(1 - e^{-\frac{t}{\tau_i}}\right), \tag{6}$$

where an ordering in terms of these time scales is assumed i.e. $\tau_0 > \tau 1 > \ldots$. Depending on the morphology, this sum can have a large number of components (**Major et al., 1993**), but in practice often only two or three components can be reliably extracted. As described in **Golowasch et al., 2009**, the slowest component $\tau_0$ is the membrane time constant and allows to infer the total capacitance of a neuron by $C = \frac{\tau_0}{R_0} = \frac{\tau_0}{v_0}I_{ext}$. In the case of an isopotential cell, the membrane time constant is the only component in the charging curve and the total capacitance can be used for the CapClamp.

### Measurement of near capacitance

For the case of two components $\tau_0, R_0$ and $\tau_1, R_1$ in the charging curve (**Equation 6**), an equivalent two compartment circuit can be identified comprising a near compartment with capacitance $C_n$ and resistance $R_n$ connected via a coupling resistance $R_a$ to a far compartment with capacitance $C_f$ and resistance $R_f$ (**Golowasch et al., 2009**). With the additional assumption of a uniform membrane time constant $\tau_m = R_n C_n = R_f C_f$, the fitted two components can be mapped to the values of these five circuit parameters, which in particular provides the near capacitance $C_n$ required for the CapClamp

$$C_n = \frac{\tau_0 \tau_1}{\tau_1 R_0 + \tau_0 R_1}. \tag{7}$$

When the capacitance is subsequently clamped to a k-fold different value, $C_t = kC_n$, the uniformity assumption has to be correspondingly adjusted to $R_n C_n = kR_f C_f$ (see "Mapping between a charging curve with two components and a two compartment circuit" in Appendix 1).

## CapClamp in dentate gyrus granule cells

### Electrophysiology

Acute brain slices were produced as described earlier (**Booker et al., 2014**). Briefly, rats were anesthetized (3% Isoflurane, Abbott, Wiesbaden, Germany) and then decapitated. Brains were removed quickly and transferred to carbogenated (95% $O_2$ / 5% $CO_2$) ice-cold sucrose-ACSF containing (in $mM$): 87 NaCl, 2.5 KCl, 25 NaHCO$_3$, 1.25 NaH$_2$PO$_4$, 25 glucose, 75 sucrose, 7 MgCl$_2$, 0.5 CaCl$_2$, 1 Na-pyruvate, 1 ascorbic acid. Horizontal brain slices of 300 μm thickness were cut using a Vibratome (VT1200 S, Leica, Wetzlar, Germany). Hippocampal tissue slices, were collected and placed in a submerged holding chamber filled with carbogenated sucrose ACSF at 32-34 °C for 30 min and then at room temperature for 15 min before recording. Experiments were alternated between left and right hemisphere slices to prevent bias due to slice condition.

For recording, slices were transferred to a submerged chamber and superfused with pre-warmed, carbogenated ACSF containing (in $mM$): 125 NaCl, 2.5 KCl, 25 NaHCO$_3$, 1.25 NaH$_2$PO$_4$, 25 glucose, 1 MgCl$_2$, 2 CaCl$_2$, 1 Na-pyruvate, 1 ascorbic acid. The bath temperature was set to 32-34 °C with a perfusion rate of 12-13 ml/min. Slices were visualized using an upright microscope (AxioScope; Zeiss) equipped with infrared differential inference contrast optics and a digital camera (Retiga EX QImaging CCD, Teledyne Photometrics, AZ, USA). Granule cells from the DG were chosen based on their anatomical location within the cell body layer as well as their morphological appearance.

Whole-cell patch-clamp electrodes were produced from borosilicate glass capillaries (outer diameter , inner diameter 1 mm, Hilgenberg, Germany) using a horizontal puller (P-97, Sutter Instruments, CA, USA) and filled with an intracellular solution consisting of (in mM): K-gluconate 130, KCl 10, HEPES 10, EGTA 10, $MgCl_2$ 2, $Na_2ATP$ 2, $Na_2GTP$ 0.3, $Na_2Creatine$ 1 and 0.1% biocytin (adjusted to pH 7.3 and 315 mOsm), giving a series resistance of 2.5-4 MΩ. All recordings were performed with a SEC LX10 amplifier (npi electronic, Germany), filtered online at 20 kHz with the built-in Bessel filter, and digitized at 20 kHz (National Instruments, UK). Following breakthrough into whole-cell configuration, we adjusted the bridge and capacitance compensation before switching to the dynamic clamp mode for recording. Cells were excluded if resting membrane potential was more depolarized than -45 mV. The liquid junction potential was not corrected.

**Table 2.** Multi-exponential fit and corresponding circuit parameters in the recorded dentate gyrus granule cells (N = 18) and a multicompartment model based on a reconstructed DGGC morphology (see "Multicompartment model of a dentate gyrus granule cell" in Methods).

|  | DGGCs (mean ± std) | Multicomp. model |
|---|---|---|
| Exp. fit |  |  |
| $\tau_0$ | 15.1±4.8 ms | 15.1 ms |
| $R_0$ | 127.1±44.6 MΩ | 119.2 MΩ |
| $\tau_1$ | 0.77±0.24 ms | 0.18 ms |
| $R_1$ | 34.5±14.7 MΩ | 12.3 MΩ |
| Circuit |  |  |
| $C_n$ | 21.0±9.4 pF | 13.0 pF |
| $R_n$ | 854.2±394.0 MΩ | 1158.0 MΩ |
| $R_a$ | 52.5±19.8 MΩ | 15.5 MΩ |
| $C_f$ | 105.8±33.0 pF | 113.7 pF |
| $R_f$ | 155.5±59.9 MΩ | 132.8 MΩ |

## Neuronal visualization and immunohistochemistry

Following recording, selected cells were immersion fixed in 4% paraformaldehyde (PFA) in 0.1 M phosphate buffer (PB, pH 7.4) at 4 °C for 24–48 hr, slices were then transferred to fresh PB. Prior to immunohistochemical processing, slices were rinsed in PB, followed by PB buffered saline (PBS, 0.9% NaCl). Slices were then rinsed in PBS and incubated in a fluorescent-conjugated streptavidin (Alexa Fluor-647, 1:1000, Invitrogen, UK) in PBS solution containing 3% NGS, 0.1% TritonX-100 and 0.05% NaN3 for 24 hr at 4 °C. Slices were rinsed in PBS and then desalted in PB before being mounted (Fluoromount-G, Southern Biotech) on 300-μm-thick metal spacers, cover-slipped, sealed, and stored at 4 °C prior to imaging.

## Confocal imaging and reconstruction

DGGCs were imaged on a laser scanning confocal microscope (FV1000, Olympus, Japan). First, a low magnification (4 x, Olympus, Japan) overview image was taken to confirm the cellular type and localization to the DG, then high resolution z-stacks were obtained with a 30x silicone oil immersion objective (N.A. 1.05, UPlanSApo, Olympus) over the whole extent of the cell (1 μm axial steps). Image stacks were stitched offline using the FIJI software package (https://imagej.net/software/fiji/imagej.net), then the cells were reconstructed and volume filled using Neutube (https://www.neutracing.com/neutracing.com) (*Feng et al., 2015*).

## Dynamic clamp setup

Data acquisition and dynamic clamp loop were controlled by RELACS, V0.9.8, RRID:SCR_017280 using a dedicated computer with a Linux-based real time operating system (https://www.rtai.org/rtai.org). The sampling frequency was set to 20 kHz and the recordings were performed in discontinuous current clamp with a duty cycle of 16.5 μs. We implemented a CapClamp procedure for RELACS that allows the user to online specify the measured capacitance $C_c$ and the desired target capacitance $C_t$ (for documentation and installation instruction, see "Data and software availability" in Appendix 1).

## Online measurement of capacitance

For the online measurement of the local capacitance, DGGCs were subjected to twenty hyperpolarizing pulses of 200 ms length with 400 ms pauses and an amplitude chosen to produce a response of −5 mV in order to minimize interference from active ionic currents. Responses were averaged and the resulting mean trajectory was fit with a sum of exponentials using the Levenberg-Marquardt method

from the python library scipy (**Virtanen et al., 2020**). Fits were performed with one, two and three components and were compared via the F-statistic (**Bardsley et al., 1986**). In all recorded DGGCs, the two component fit was significantly better than the one exponential fit ($p < 0.05$, 18/18), whereas no cell exhibited a significant third component ($p < 0.05$, 0/18). Finally, the extracted two components were mapped to a two compartment circuit as explained above and the near capacitance was then used in the subsequent CapClamp (**Table 2**).

An offline reexamination revealed that in several recorded cells the above fitting procedure yielded inaccurate estimates of the exponential components, e.g. very short fast components due to an artefactual voltage dip *before* pulse onset. To circumvent these problems, improved offline fits were performed for the artifact-free recharging at the pulse end ( see "Adapted fitting procedure of dentate gyrus charging curves" in Appendix 1). In 8/18 cells, the offline and the original online estimate of the near capacitance differed by less than 20%, but overall the offline measurement yielded higher capacitance values than originally used for the CapClamp (offline: $21.0 \pm 9.4\,\mathrm{pF}$/, online: $14.9 \pm 4.8\,\mathrm{pF}$/). In contrast to the online measurement, the offline procedure reported a better fit with three components for a subset of cells ($p < 0.05$, 7/18), but for the analysis presented here the result of the two component fit is used in all cells.

## Protocol 1: Verification of altered capacitance

After online measurement of the capacitance, each DGGC was clamped at a range of capacitances from 60% to 300% of the original near capacitance. For each clamped capacitance, the above offline capacitance measurement protocol was repeated to see how the CapClamp altered the slow and fast components. These time scale and amplitude changes were then mapped to the corresponding two compartment circuit parameters to compare them to the target capacitance (see Measurement of near capacitance). Due to the difference between online and offline estimate of the original near capacitance, we corrected the original target capacitance to $C_t^{\mathrm{corr}} = C_c^{\mathrm{off}} + \Delta C_t$, which preserves the targeted capacitance change $\Delta C_t = C_t - C_c^{\mathrm{on}}$. Equally, the clamping factors in the mapping were updated to $k = \frac{C_t^{\mathrm{corr}}}{C_c^{\mathrm{off}}}$.

## Protocol 2: Analysis of f-I curves and spike shapes

In a subset of cells, after measuring near capacitance, an fI curve was obtained for the original capacitance and for target capacitances in the above range. Current pulses were 1 s long and repeated three times, at amplitudes ranging from 90% to 200% of an estimated rheobase. This rheobase was estimated by the first occurrence of spiking in response to a ramp (length: 5 s, height: 250 pA). For a quantitative comparison, the resulting fI curves were fit by a square-root function

$$f(I) = \Theta(I - I_{\mathrm{rheo}})\mathrm{gain}\sqrt{I - I_{\mathrm{rheo}}} \tag{8}$$

which captured their type 1 firing with a continuous frequency-current relationship (**Izhikevich, 2006**, p. 168). Cells with more than 30% varying input resistance within the protocol and/or a non-monotonically increasing fI curves were excluded from the analysis.

Spikes were detected as a minimum 10 mV elevation over the average depolarization during the pulse. For the mean action potential (AP) shape, varying spike forms from the initial (< 300 ms) part of the pulse were discarded. The extracted AP features were peak amplitude, threshold voltage and threshold time to peak (voltage derivative crossing 10 mV/ms), height (difference between peak and threshold), temporal width at half of the height and fast afterhyperpolarization (fAHP; a voltage dip of –0.5 mV or larger within 10 ms after the spike). For threshold and fAHP detection, the spike shape was filtered with a digital 4th order Butterworth filter with critical frequencies 3.3 kHz, respectively 1 kHz.

To detect, whether changes in capacitance affect the action potential generating currents, we compared the recorded spikes with hypothetical ones obtained by assuming unaltered currents with respect to the original near capacitance. For a target capacitance $C_t$, such a hypothetical spike would be a scaled version of the original spike,

$$V_{hypo}(t) = V_c(t_0) + \frac{C_c}{C_t}\left(V_c(t) - V_c(t_0)\right),$$

where $V_c(t)$ is the spike form at the original cell capacitance $C_c$ and the initial time $t_0$ was chosen to be $t_{spike} - 3$ms short before onset of the spike generating currents. Changes in the measured spike shape compared to this hypothetical shape signal a change of the underlying currents.

## Simulations of the CapClamp

Simulations of neuron models coupled to the CapClamp were implemented using the neuron simulator Brian2 (*Stimberg et al., 2019*) and the CapClamp was realized using the Brian2 provided NetworkOperation that updated the clamp current every sampling interval using *Equation 5* with zero delay between voltage sampling and current injection (for links to the available code, see "Data and software availability" in Appendix 1).

### Biophysical neuron model

In order to test the CapClamp in the presence of active ionic conductances, a Wang-Buzsáki (WB) neuron, a single compartment model of hippocampal interneurons, was used (*Wang and Buzsáki, 1996*). Gating dynamics and peak conductances of the transient sodium current and the delayed rectifier potassium current were modeled as described earlier (*Hesse et al., 2017*, Appendix A). The specific membrane capacitance was chosen as $C_m = 0.75 \frac{\mu\mathrm{F}}{\mathrm{cm}^2}$ and the membrane area was set to $A = 20000 \, \mu m^2$, so that the original cell capacitance was 150 pF. When the capacitance is varied, the WB neuron undergoes a well-characterized series of bifurcations; in particular it exhibits a saddle-node loop (SNL) bifurcation at $C_m = 1.47 \frac{\mu\mathrm{F}}{\mathrm{cm}^2}$ accompanied by an abrupt doubling of the firing rate (*Hesse et al., 2017*). For the demonstration of the CapClamp here, we decided to restrict the tested capacitances to the regime below this critical value, but we confirmed via additional simulations that the CapClamp continues to work beyond the bifurcation (data not shown).

Simulations were performed with the second order Runge-Kutta method, a time step of 1 $\mu$s and dynamic clamp loop frequencies up to 100 kHz. Analysis of spike shapes and f-I curves was performed in the same way as for the recorded cells.

### Analytically expected effect of capacitance on the form of the f-I curve

How the form of the f-I curve depends on capacitance can be analytically calculated for a single-compartment conductance-based neuron model undergoing a saddle-node on a limit cycle bifurcation at spiking onset like the WB model considered here (*Izhikevich, 2006*), pp. 162–168; (*Schleimer and Schreiber, 2018*). In this case, the time between two spikes $T_{\mathrm{isi}}$ is dominated by the slow traversal $T_2$ of the saddle node, which close to threshold is multiple times longer than the brief duration $T_1$ of the spike and can be derived by considering solely local dynamics

$$T_{\mathrm{isi}} = T_1 + T_2 \approx T_2 = \frac{\pi}{\sqrt{ac\left(I - I_{\mathrm{rheo}}\right)}}$$

where $a$ and $c$ parametrize the normal form of the dynamics around the saddle node and $I_{\mathrm{rheo}} = I_{\mathrm{sn}}$ is the current value where the saddle node bifurcation occurs. Inverting the inter spike interval to get the frequency then gives the square root form of the f-I curve (see *Equation 8*). Under the assumption of fast gating kinetics, the $\mathrm{gain} = \frac{\sqrt{ac}}{\pi}$ is expected to be proportional to the inverse of the capacitance $\frac{1}{C}$, because the relevant time scale for the local slow dynamics around the saddle node is the membrane time constant implying that the traversal duration scales as $T_2 \propto \tau \propto C$. The rheobase current in contrast is expected to remain constant, because equilibrium points are independent of the time scales of the dynamics. Formally calculating the normal form parameters $a$ and $c$ confirms these expectations (see "Formal derivation of f-I curve gain and rheobase dependence oncapacitance" in Appendix 1).

The $\frac{1}{C}$ dependence of the gain allows to estimate an expected gain reduction for small capacitance changes around the original capacitance $C_c$ by Taylor expansion

$$\Delta \mathrm{gain} = \mathrm{gain}(C_c) - \mathrm{gain}(C_c + \Delta C) \approx -\frac{\mathrm{gain}(C_c)}{C_c} \Delta C$$

which we compare for both simulated neuron and DGGCs to the observed gain reduction.

## Multicompartment model of a dentate gyrus granule cell

For a controlled test of the CapClamp in an electrotonically non-compact cell, a morphologically reconstruction of a recorded DGGC was used as the basis for a multicompartment simulation. Soma and the two dendritic trees had a total area of 14,126 $\mu m^2$. The axon was removed for the simulation. Membrane properties were assumed to be uniform and chosen such that they reproduced the average values of the total capacitance and the membrane time constant observed in the experiments: $C_m = \frac{\overline{C_n + C_f}}{A} \approx 0.9\,\frac{\mu F}{cm^2}$ and $R_m = \frac{\tau_0}{C_m} \approx 16800\,\Omega cm^2$. The axial resistivity was chosen as $R_{axial} = 300\,\Omega cm$. Simulations were performed with exponential Euler integration, a time step of 10 $\mu$s and a dynamic clamp sampling frequency of 20 kHz. Capacitance measurement and clamp procedure were the same as in the recorded DGGCs (*Table 2*).

## Acknowledgements

We thank Jan Benda and Lukas Sonnenberg for their dedicated support with dynamic clamp and fruitful discussions. We are grateful to Eve Marder and Ekaterina Morozova for being able to test the RTXI implementation of the CapClamp and apply the CapClamp in neurons of the crustacean stomatogastric ganglion. We thank Robert Gowers and Philipp Norton for valuable feedback on the manuscript.

This project has received funding from the European Research Council (ERC) under the European Union's Horizon 2020 research and innovation program (grant agreement No 864243). The article processing charge was funded by the Deutsche Forschungsgemeinschaft (DFG, German Research Foundation) – 491192747 and the Open Access Publication Fund of Humboldt-Universität zu Berlin.

## Additional information

### Funding

| Funder | Grant reference number | Author |
| --- | --- | --- |
| Bundesministerium für Bildung und Forschung | 01GQ1403 | Jan-Hendrik Schleimer Susanne Schreiber |
| Deutsche Forschungsgemeinschaft | GRK 1589/2 | Paul Pfeiffer Federico José Barreda Tomás |
| Deutsche Forschungsgemeinschaft | EXC 257 | Federico José Barreda Tomás Imre Vida |
| Deutsche Forschungsgemeinschaft | FOR 2134 | Federico José Barreda Tomás Imre Vida |
| H2020 European Research Council | 864243 | Susanne Schreiber |
| Einstein Stiftung Berlin | EZ-2014-224 | Jiameng Wu |

The funders had no role in study design, data collection and interpretation, or the decision to submit the work for publication.

### Author contributions

Paul Pfeiffer, Conceptualization, Data curation, Formal analysis, Investigation, Methodology, Project administration, Software, Visualization, Writing – original draft, Writing – review and editing; Federico José Barreda Tomás, Data curation, Investigation, Methodology, Software, Validation, Visualization, Writing – review and editing; Jiameng Wu, Formal analysis, Investigation, Methodology, Software, Visualization, Writing – review and editing; Jan-Hendrik Schleimer, Conceptualization, Formal analysis, Methodology, Supervision, Writing – review and editing; Imre Vida, Funding acquisition, Project administration, Resources, Supervision, Writing – review and editing; Susanne Schreiber, Conceptualization, Funding acquisition, Project administration, Resources, Supervision, Writing – original draft, Writing – review and editing

## Author ORCIDs
Paul Pfeiffer http://orcid.org/0000-0001-5324-5886
Jiameng Wu http://orcid.org/0000-0002-6266-7666
Jan-Hendrik Schleimer http://orcid.org/0000-0002-2156-330X
Imre Vida http://orcid.org/0000-0003-3214-2233
Susanne Schreiber http://orcid.org/0000-0003-3913-5650

## Ethics

All procedures and animal maintenance were performaed in accordance with institutional guidelines, the German Animal Welfare Act, the European Council Directive 86/609/EEC regarding the protection of animals, and guidelines from local authorities (Berlin, T-0215/11).

## Decision letter and Author response

Decision letter https://doi.org/10.7554/eLife.75517.sa1
Author response https://doi.org/10.7554/eLife.75517.sa2

---

# Additional files

## Supplementary files
• Transparent reporting form

## Data availability

All data generated, analysis code as well as computational modelling code is uploaded on https://zenodo.org/, see article section Data and software availability.

The following dataset was generated:

| Author(s) | Year | Dataset title | Dataset URL | Database and Identifier |
|---|---|---|---|---|
| Pfeiffer P, Barreda Tomás F J | 2021 | Capacitance Clamp Demonstration in Rat Dentate Gyrus Granule Cells | https://doi.org/10.5281/zenodo.5552207 | Zenodo, 10.5281/zenodo.5552207 |

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

# Appendix 1

## Impedance of a capacitance-clamped RC circuit

The impedance of a cell captures its linear response to the whole range of input frequencies (see *Figure 1—figure supplement 1b*). In the following, we derive the impedance of a passive membrane, an RC circuit, with capacitance $C_c$ coupled to the CapClamp and compare it to the impedance of an RC circuit with the target capacitance $C_t$.

### Analysis of the dynamic clamp via the Z-transform

In general, the dynamic clamp technique forms a digital filter, mapping the incoming sampled voltages to injected currents. For a sampling interval $\Delta t$, a linear mapping such as the CapClamp has the form

$$I_{\mathrm{dyn}}(i\Delta t) = \sum_{j=0}^{N} \nu_j V\left((i-j)\Delta t\right) + \sum_{k=1}^{M} \gamma_k I_{\mathrm{dyn}}\left((i-k)\Delta t\right), \tag{9}$$

where $N$ and $M$ determine history of voltage and current values, respectively, taken into account. For the CapClamp, the coefficients depend on cell capacitance $C_c$, target capacitance $C_t$ and the sampling interval (see *Equation 5*),

$$
\begin{aligned}
\nu_0 &= \frac{C_C - C_t}{C_t}\frac{C_C}{\Delta_t}, \\
\nu_1 &= -\nu_0, \\
\gamma_1 &= -\frac{C_C - C_t}{C_t}.
\end{aligned}
\tag{10}
$$

This linear mapping can be represented and analyzed using the Z-transform (*Dorf and Bishop, 2010*, Ch. 13),

$$\hat{I}(z) = F_{\mathrm{dyn}}(z)\hat{V}(z), \tag{11}$$

$$F_{\mathrm{dyn}}(z) = \frac{\sum_{j=0}^{N} \nu_j z^{-j}}{1 - \sum_{k=1}^{M} \gamma_k z^{-k}}. \tag{12}$$

where the transfer function follows from the properties of the Z-transform: linearity $\lambda X_i \xrightarrow{Z} \lambda \hat{X}(z)$ and delay transformation $X_{i-1} \xrightarrow{Z} z^{-1}\hat{X}(z)$ (*Dorf and Bishop, 2010*, Table 13.2),

If the cell also forms a linear system, like the RC circuit, the transfer function of the coupled system *Appendix 1—figure 1* is given by *Dorf and Bishop, 2010*, Table 2.6.

$$H_{\mathrm{cell+dyn}}(z) = \frac{H_{\mathrm{cell}}(z)}{1 - H_{\mathrm{cell}}(z)F_{\mathrm{dyn}}(z)}, \tag{13}$$

where $H_{\mathrm{cell}}(z)$ is the Z-transform of the membrane filter, e.g. $H_{\mathrm{cell}}(z) = H_{\mathrm{RC}}(z)$.

The transfer function of the coupled system $H_{\mathrm{cell+dyn}}(z)$ can then be compared with the one of the target system $H_{\mathrm{target}}(z)$ (*Appendix 1—figure 1*). Additionally, the frequency-dependent impedance can be retrieved from the transfer function by

$$Z_{\mathrm{cell+dyn}}(f) = H_{\mathrm{cell+dyn}}(e^{i2\pi f \Delta t}). \tag{14}$$

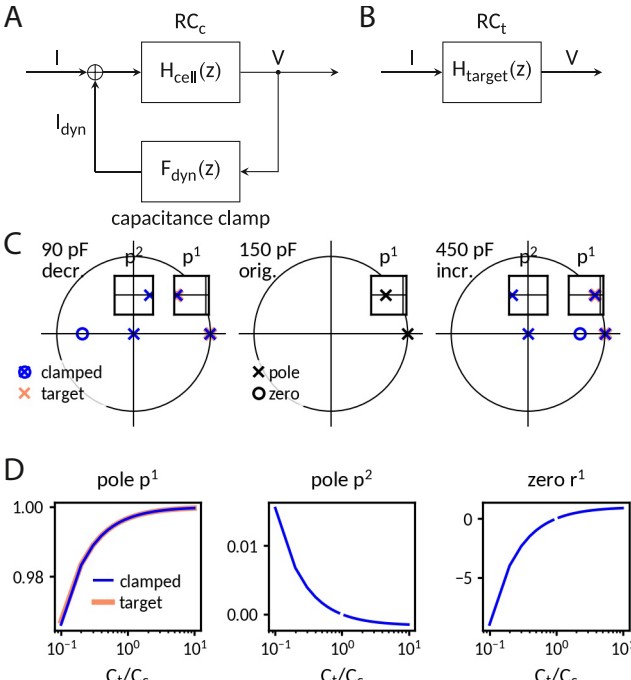

**Appendix 1—figure 1.** Analysis of the capacitance clamp as a discrete feedback filter. (**A**) Block diagram of the coupled system: RC circuit with original capacitance Cc and capacitance clamp feedback current. (**B**) Block diagram of the target system: RC circuit with target capacitance Ct. (**C**) Pole-zero plot of the transfer functions at a decreased (left), the original and an increased capacitance. In addition to mimicking the pole of the target system, the clamped system has an additional pole and an additional zero. (**D**) Pole and zero position versus capacitance.

## Transfer function of the CapClamp

The Z-transform of the CapClamp filter can be read directly from the general form of the transfer function (**Equation 12**) and the CapClamp feedback coefficients (**Equation 10**),

$$F_{\text{dyn}}(z) = \frac{C_c - C_t}{C_t} \frac{C_c}{\Delta t} \frac{1 - z^{-1}}{1 + \frac{C_c - C_t}{C_t} z^{-1}}. \tag{15}$$

## Transfer function of the RC circuit

In an RC circuit, the dynamics of the voltage are

$$C \frac{\text{d}V}{\text{d}t} = -\frac{V}{R} + I.$$

Thus, in a single time step $\Delta t$, when the current is fixed, the voltage evolves as

$$V(k\Delta t) = V((k-1)\Delta t) e^{-\frac{\Delta t}{\tau}} + RI(1 - e^{-\frac{\Delta t}{\tau}}),$$

where $\tau = RC$ is the time constant. Applying the Z-transform results in the transfer function

$$H_{RC}(z) = R\left(1 - e^{-\frac{\Delta t}{\tau}}\right) \frac{1}{z - e^{-\frac{\Delta t}{\tau}}}, \tag{16}$$

which is subsequently used as the cell's transfer function $H_{\text{cell}}(z) = H_{RC}(z)$.

## Transfer function of the clamped RC circuit

Introducing $K = \frac{C_c - C_t}{C_t}$ and $h_c = \frac{\Delta t}{\tau_c}$, the RC circuit (**Equation 16**) and CapClamp (**Equation 15**) transfer functions can be combined using **Equation 13** to get the transfer function of the combined system

$$H_{\text{cell+dyn}}(z) = R(1 - e^{-h_c}) \frac{z + K}{z^2 + (K - e^{-h_c} - \frac{1}{h_c} K(1 - e^{-h_c}))z - K(e^{-h_c} - \frac{1}{h_c}(1 - e^{-h_c}))}. \tag{17}$$

In comparison, the transfer function of the target RC circuit reads

$$H_{\text{target}}(z) = R \left( 1 - e^{-h_t} \right) \frac{1}{z - e^{-h_t}}$$

with $h_t = \frac{\Delta t}{\tau_t} = \frac{\Delta t}{RC_t}$ reflecting the different target capacitance.

*Figure 1—figure supplement 1b* compares the resulting impedances for decreased and increased capacitances. As discussed in the "Results" section , the impedance amplitudes fit well up to a tenth of the dynamic clamp frequency. A closer look at the transfer function explains the fit at low frequencies and the deviations at higher frequencies.

## Input resistance is preserved

The input resistance is equal to the impedance at zero frequency, that is at $z = e^{i2\pi 0} = 1$, which for both coupled and target system is the original resistance,

$$H_{\text{cell+dyn}}(1) = H_{\text{target}}(1) = R. \tag{18}$$

## Poles and zeros

For a further comparison, poles and zeros of the transfer functions are calculated. To simplify the expressions, it is assumed that the time constant of the original and target circuits are much larger than the sampling interval, that is $h_c \ll 1$ and $h_t \ll 1$.

## Target circuit

The target circuit has no zero and a single pole located at

$$p_t^{(1)} = e^{-h_t} = 1 - h_t + \dots \tag{19}$$

## Capacitance clamped circuit

The coupled system has one zero at

$$r_c^{(1)} = -K = 1 - \frac{C_c}{C_t}. \tag{20}$$

The clamped circuit has two poles at

$$p_c^{(1)} = 1 - (1 + K)h_c + \dots \tag{21}$$

and

$$p_c^{(2)} = \frac{K}{2}h_c + \dots \tag{22}$$

## Comparison of poles

All poles and zeros for an RC circuit in its original state and clamped at decreased and increased capacitances are shown in *Appendix 1—figure 1*. The first pole of the clamped circuit coincides with the one of the target circuit: $p_c^{(1)} = 1 - \frac{C_c}{C_t} \frac{\Delta t}{RC_c} = 1 - h_t = p_t^{(1)}$. As these pole lies close to $z = 1$, they determine the lower frequency response, which explains why the impedance amplitudes fit so well in this range.

In addition to moving the existent pole of the cell circuit to the one of the target circuit, the CapClamp creates an additional pole $p_c^{(2)} \approx \frac{h_c}{2}(\frac{C_c}{C_t} - 1)$ and a new zero $r_c^{(1)} = 1 - \frac{C_c}{C_t}$. Thus, at an increased capacitance $C_t > C_c$, the new pole lies in the left half of the unit circle and thereby increases the impedance at higher frequencies. In contrast, at a decreased capacitance, the additional zero moves into the left half of the complex plane and thereby decreases the impedance at higher frequencies.

## Stability

For the investigated RC circuit with $R$=100 MΩ and $C$ = 150 pF and a sampling interval of 50 us, both poles of the capacitance clamped system remain within the unit circle (*Appendix 1—figure 1*) for the tested range from 0.1 to 10 times the original capacitance. As the coupled system is naturally causal, this implies that the transfer function of the clamped circuit is stable for this range of target capacitances, i.e. there are no unstable oscillations.

## Mapping between a charging curve with two components and a two compartment circuit

In the following, we explain how a charging curve of a cell with two components can be mapped to the parameters of a two compartment circuit, which we used to extract the local capacitance in the recorded dentate gyrus granule cells (see *Figure 3*). We first report the approach and results derived earlier (*Golowasch et al., 2009*) and then explain how to extend the mapping when the capacitance is clamped to a modified value.

Golowasch et al. derived expressions for the near capacitance and the other circuit parameters by comparing the impedance of a two compartment circuit in *Figure 3A*

$$Z(s) = \frac{1}{\frac{1}{R_n} + sC_n + \frac{1}{R_a + \frac{1}{\frac{1}{R_f} + sC_f}}}$$

(23)

with the impedance of a system whose response to a step currents is a sum of two exponentials

$$Z(s) = R_0 \frac{1}{1 + s\tau_0} + R_1 \frac{1}{1 + s\tau_1}.$$

(24)

The comparison of these two impedances gives four equations linking the circuit parameters and the two components of the charging curve:

$$R_0 + R_1 = \frac{R_a R_n + R_f R_n}{R_a + R_f + R_n},$$

(25)

$$R_0\tau_0 + R_1\tau_1 = \frac{R_a R_n R_f C_f}{R_a + R_f + R_n},$$

(26)

$$\tau_0 + \tau_1 = \frac{(R_a + R_n)C_f R_f + (R_a + R_f)C_n R_n}{R_a + R_f + R_n},$$

(27)

$$\tau_0\tau_1 = \frac{R_a C_n R_n C_f R_f}{R_a + R_f + R_n}.$$

(28)

To solve this set of equations, they assume that the membrane time constant is the same in all compartments $C_n R_n = C_f R_f = \tau_c$. However in a clamped neuron, where the near capacitance is targeted to be modified to a k-fold different value, this equation becomes

$$C_n R_n = kC_f R_f,$$

(29)

where $k = \frac{C_{n,clam.}}{C_{n,orig.}}$.

For the unclamped case, $k = 1$, the mapping from the two components to the circuit parameters is

$$R_n = R_0 + \frac{\tau_0}{\tau_1}R_1,$$

(30)

$$C_n = \frac{\tau_0\tau_1}{\tau_1 R_0 + \tau_0 R_1},$$

(31)

$$R_f = \frac{R_0\tau_1}{R_1\tau_0}\left(R_0 + \frac{\tau_0}{\tau_1}R_1\right)$$

(32)

$$C_f = \frac{R_1\tau_0}{R_0\tau_1}\frac{\tau_0\tau_1}{\tau_1 R_0 + \tau_0 R_1}$$

(33)

$$R_a = \frac{\tau_1}{\tau_0 - \tau_1}\left(R_0 + \frac{\tau_0}{\tau_1}R_1\right)\left(1 + \frac{R_0\tau_1}{R_1\tau_0}\right).$$

(34)

For the clamped case, $k \neq 1$, we used the python package sympy to solve the equations.

## Adapted fitting procedure of dentate gyrus charging curves

The initial online capacitance measurement was based on fitting the charging curve at the beginning of the current pulse. Posterior analysis showed an artefactual voltage drop of –0.2 mV starting about 0.2ms before pulse onset (probably due to coupling of the DAQ measurement card and the motherboard of the dynamic clamp computer), which limited the reliability of the online fit for cells with a small fast component. As no such artifact was observed for the recharging at the end of the pulse, this part was used in an improved offline fit. Additional measures to improve the fit were: cut of the first 0.2ms after pulse end to minimize electrode artifacts, limiting the fit to the first 60 ms (3–4 times $\tau_0$) after the pulse to prioritize the early part of the charging curve and a switch to the python package lmfit for better evaluation of parameter confidence bounds (https://lmfit.github.io/lmfit-py/). Furthermore, the finite rise time of the current injection by the amplifier was taken into account by adapting the original form of the charging curve (**Equation 6**) to

$$V(t) = I_{ext} \left[ \sum_{i;\tau_i \neq \tau_a} \frac{R_i}{\tau_i - \tau_a} \left( \tau_i \left( 1 - e^{-\frac{t}{\tau_i}} \right) - \tau_a \left( 1 - e^{-\frac{t}{\tau_a}} \right) \right) + \sum_{i;\tau_i = \tau_a} R_i \left( 1 - e^{-\frac{t}{\tau_i}} - \frac{t}{\tau_i} e^{-\frac{t}{\tau_i}} \right) \right],$$

(35)

where the current rise time of the amplifier $\tau_a$ ($87 \pm 2\,\mu s$) was obtained by fitting the recorded injected current for the current step command by a simple exponential. A comparison of the two exponential components and the resulting circuit parameters for the online and offline fitting procedures is show in **Appendix 1—table 1**.

For the charging curves under capacitance clamp, the fitting procedure for the charging curve with two exponentials was initialized with values as expected for the targeted capacitance change: mapping the fitting results of the unclamped response to a two compartment circuit, changing the near capacitance to the targeted value and finally mapping this altered circuit back to the expected time scale and amplitudes. This initialization improved the fits especially at increased near capacitances, where the amplitude of the fast component becomes smaller.

## Formal derivation of f-I curve gain and rheobase dependence on capacitance

To confirm the expectations of gain and rheobase dependence on capacitance in a single compartment neuron model (see "Analytically expected effect of capacitance on the form of the f-I curve" in Methods), we here sketch the calculation of the normal form parameters $a$ and $c$ following *Izhikevich, 2006*, pp. 162–163. In principle, $a$ and $c$ can be calculated for arbitrary gating kinetics by projecting the dynamics on the center manifold (*Schleimer and Schreiber, 2018*). For the assumption

**Appendix 1—table 1.** Comparison of online and offline fits to charging curves in the recorded dentate gyrus granule cells (N = 18).

|  | Online fit (mean ± std) | Offline fit (mean ± std) |
| --- | --- | --- |
| Two comp. |  |  |
| $\tau_0$ | 14.9±4.8 ms | 15.1±4.8 ms |
| $R_0$ | 136.9±47.5 MΩ | 127.1±44.6 MΩ |
| $\tau_1$ | 0.41±0.23 ms | 0.77±0.24 ms |
| $R_1$ | 25.1±14.1 MΩ | 34.5±14.7 MΩ |
| Circuit |  |  |
| $C_n$ | 14.9±4.7 pF | 21.0±9.4 pF |
| $R_n$ | 1106.3±519.3 MΩ | 854.2±394.0 MΩ |
| $R_a$ | 34.9±19.9 MΩ | 52.5±19.8 MΩ |
| $C_f$ | 99.1±33.7 pF | 105.8±33.0 pF |
| $R_f$ | 159.6±58.1 MΩ | 155.5±59.9 MΩ |

of small gating time constants, however, they can be expressed in simpler terms using the steady state I-V relation of the neuron divided by its membrane capacitance

$$\mathbf{I}(V, I) = \frac{1}{C}\left(I - I_\infty(V)\right) = \frac{1}{C}\left(I - \sum_{i=1}^{N} g_i \prod_{j=1}^{M_i} x_{ij,\infty}(V)^{p_{ij}}(V - E_i)\right)$$

taking the form $a = \frac{1}{2}\frac{\partial^2 \mathbf{I}(V,I)}{\partial V^2}\Big|_{V=V_{sn}, I=I_{sn}}$ and $c = \frac{\partial \mathbf{I}(V,I)}{\partial I}\Big|_{V=V_{sn}, I=I_{sn}}$, where the saddle node voltage $V_{sn}$ and current $I_{sn}$ are given by the equations

$$\mathbf{I}(V, I)|_{V=V_{sn}, I=I_{sn}} = 0,$$

$$\frac{\partial \mathbf{I}(V,I)}{\partial V}\Big|_{V=V_{sn}, I=I_{sn}} = 0.$$

In summary, both $a$ and $c$ are proportional to $\frac{1}{C}$ and the rheobase current $I_{rheo} = I_{sn} = I_\infty(V_{sn})$ is independent of capacitance, thus confirming the expected scaling.

## Data and software availability

- Electrophysiological recordings of capacitance clamped dentate gyrus granule cells: Pfeiffer, P., & Tomás F. J. B. (2021). Capacitance clamp demonstration in rat dentate gyrus granule cells. https://doi.org/10.5281/zenodo.5552207
- Project repository with capacitance clamp module for https://scicrunch.org/resolver/RRID: SCR_017280RELACS and custom analysis/simulation in python: Pfeiffer, P., Tomás, F. J. B., Wu, J., Schleimer, J.-H., Vida, I., & Schreiber, S. (2021). Software for: A dynamic clamp protocol to artificially modify cell capacitance. https://doi.org/10.5281/zenodo.6322768
- Capacitance clamp plugin for http://rtxi.org/RTXI, a real-time data-acquistion and control application for biological research that allows to extend a conventional electrophysiology setup for dynamic clamp experiments (**Patel et al., 2017**). Capacitance_clamp_rtxi_module: https://doi.org/10.5281/zenodo.5553946

