## [Editor Report]

The manuscript introduces a new enhancement to the dynamic clamp technique, CapClamp that, analogous to the artificial conductances of standard Dynamic Clamp, allows the experimenter to adjust the somatic time constant by setting a new membrane artificial capacitance independent of any change in input resistance. The technique is shown to have application for studying temporal integration, energetic costs of spiking and bifurcations. The technique is rigorously tested in model and physiological application and is robust when sampling frequency of the feedback (clamp) loop is fast compared to the fastest electrical event in a neuron (usually action potentials), and for vertebrate neurons it should be 20KHz or faster and yet faster for fast spiking neurons.

---

## [Decision Letter]

**Decision letter after peer review:**

Thank you for submitting your article "A dynamic clamp protocol to artificially modify cell capacitance" for consideration by *eLife*. Your article has been reviewed by 3 peer reviewers, including Ronald L Calabrese as the Senior and Reviewing Editor and Reviewer #3. The following individual involved in review of your submission has agreed to reveal their identity: Jorge Golowasch (Reviewer #1).

Essential revisions:

1) As discussed by Reviewer #2 in their public review, the changes in capacitance have a weak effect on excitability in real neurons. The authors should definitely apply the same comparison already performed for action potential, i.e., expected changes vs. real changes in FI curve following capacitance clamp. Maybe in DGGCs, not much effect is expected on gain, but this has to be clearly demonstrated. Otherwise, it raises concerns about the efficiency of capacitance clamp beyond the soma.

2) Please address all the concerns expressed in the Recommendations for the authors.

*Reviewer #1 (Recommendations for the authors):*

This is an excellent paper. The new dynamic clamp method described here to control membrane capacitance is based on sound theory, it is well described and tested. I have no major concerns. In fact, I appreciate the authors for developing this in such a timely manner for me, as I was planning to do something similar myself.

*Reviewer #2 (Recommendations for the authors):*

1) The authors demonstrate in the neuron model that manipulating capacitance not only affects action potential waveform but also significantly alters the excitability profile of the cell, modifying in particular the gain of the fI curve of the neuron. However, when tested on DGGCs, the effect of capacitance on action potential shape is very strong but the effect on excitability is very mild. The overall change in gain is close to 20% for a 5-fold change in capacitance, while a similar change in capacitance induced a ~2-fold in gain in the neuron model. While the authors say that the results in real neurons are similar to the ones obtained in the simulated neuron, the quantitative difference is large enough to contradict that statement. Moreover, this discrepancy questions the ability of the capacitance clamp to efficiently modify capacitance in real neurons. In fact, concerning action potential shape, the authors compare the effect of capacitance manipulation in real neurons with the expected effect (Figure 4B), but do not present this comparison for excitability measurements. It would have been very interesting to see whether the actual results significantly depart from the expected results.

2) Concerning the potential impact of capacitance clamp, and since most changes in capacitance in physiological contexts seem to be related to neuronal growth, it would have been really interesting to compare the impact of manipulating capacitance with the impact of manipulating concomitantly input resistance and capacitance, which is expected when neuronal size is changing, for instance during development. This type of comparison would also help underlining the significant contribution of changes in capacitance, which have been so far undermined. Testing these two manipulations in parallel would greatly help to disentangle the specific contributions of changes in membrane resistance and capacitance during neuronal growth, and would emphasize the value of the capacitance clamp tool.

*Reviewer #3 (Recommendations for the authors):*

The manuscript is very clearly written and well-focused.

Lines 93-94: "…change with respect to the original capacitance (e.g. Ct=67.4 pF: C=67.5 pF; Ct=336.9 pF: C=338.1 pF), whereas the… Why are these examples chosen and not a more uniform range?

Section 2.5.1: I found this very confusing. In an RC circuit the response sizes to a sequence of current pulses as measured by baseline to peak are identical. It is very confusing to say "…the cell's response to the second one should be higher than to the first one." Or to say "…as apparent by the larger step sizes in the stair-like voltage response and the finally higher ratio of last to first pulse response." Please rewrite.

Section 2.5.3: This section was the only part of the paper I found unconvincing. It is a foregone conclusion that the CapClamp can find the critical capacitance in the Wang-Buzsáki neuron, given Section 2.3. The failure of the technique to find the critical capacitance in a dentate gyrus granule cell is thus an ambiguous result. Is this a technical failure or a real result? I suggest deleting this section.

---

## [Author Response]

Essential revisions:1) As discussed by Reviewer #2 in their public review, the changes in capacitance have a weak effect on excitability in real neurons. The authors should definitely apply the same comparison already performed for action potential, i.e., expected changes vs. real changes in FI curve following capacitance clamp. Maybe in DGGCs, not much effect is expected on gain, but this has to be clearly demonstrated. Otherwise, it raises concerns about the efficiency of capacitance clamp beyond the soma.

We thank the reviewers for pointing out this apparent discrepancy between model and experiment. The discrepancy, however, is not as strong as the text may have suggested. In the revised manuscript, we now provide a quantitative analysis of the effects on excitability in the simulated neuron models and the dentate granule cells (DGGCs) as well as a comparison to expected changes based on theoretical predictions for interspike interval duration (for details see the full answer to Recommendation 1 of reviewer 2 on page 6). In summary, this re-analysis shows that the discrepancy is actually smaller than suggested by the comparison in terms of relative capacitance changes. Capacitance effects on DGGC excitability in terms of gain reduction are reduced by a factor of 0.5 in comparison to those in the model. This remaining discrepancy is consistent with theoretical predictions – less excitable cells should also exhibit a smaller gain reduction, when capacitance is increased. The additional gain analysis is thus further evidence that in the recorded DGGCs the somatic capacitance clamp was sufficient to effectively change the capacitive load of the spike generating compartment. Thus, we are confident that the CapClamp also provides this control in other cell types with spike initiation at or close to the soma.

2) Please address all the concerns expressed in the Recommendations for the authors.

We thank all reviewers for the thorough and helpful comments and addressed all concerns expressed in the Recommendations.

Reviewer #2 (Recommendations for the authors):1) The authors demonstrate in the neuron model that manipulating capacitance not only affects action potential waveform but also significantly alters the excitability profile of the cell, modifying in particular the gain of the fI curve of the neuron. However, when tested on DGGCs, the effect of capacitance on action potential shape is very strong but the effect on excitability is very mild. The overall change in gain is close to 20% for a 5-fold change in capacitance, while a similar change in capacitance induced a ~2-fold in gain in the neuron model. While the authors say that the results in real neurons are similar to the ones obtained in the simulated neuron, the quantitative difference is large enough to contradict that statement. Moreover, this discrepancy questions the ability of the capacitance clamp to efficiently modify capacitance in real neurons. In fact, concerning action potential shape, the authors compare the effect of capacitance manipulation in real neurons with the expected effect (Figure 4B), but do not present this comparison for excitability measurements. It would have been very interesting to see whether the actual results significantly depart from the expected results.

We thank the reviewer for pointing out the apparent discrepancy of gain dependence on capacitance between model and experiment. This discrepancy, however, is actually smaller than it appeared in the original manuscript. As Figure 4 on the DGGCs emphasized the near capacitance, which the CapClamp modifies, our presentation suggested comparing gains in terms of relative changes of near capacitance in the experiment and total capacitance in the model. These relative changes of the modified capacitance, however, are not the correct base for a comparison, because in the case of the DGGCs they do not take into account the large constant (far) dendritic capacitance.

In the revised manuscript, we therefore provide an explicit comparison between the model and experiment in terms of absolute capacitance changes, i.e., for the model C_t_-C_cell_ (Figure 2 E and page 7) and for the DGGCs C_t_-C_near_ (Figure 4 D and pages 12-13). In other words, we report how much the fI curve gain changes per a 10 pF increase of capacitance. In addition, we use an analytical argument to discuss how much the gain is expected to change. Our analysis shows that the dependence of gain on capacitance in the DGGCs is weaker by a factor of ~0.5 in comparison to that of the model. Such a remaining difference is expected, because the DGGCs have an overall smaller gain (for the detailed argument see below). An additional factor might be the degree of electric coupling of somatic and axonal compartment, which can vary among cells. Yet, a significant gain reduction in the majority of DGGCs – within the expected range – shows that somatic capacitance control was sufficient to affect spike generation. Overall, we are confident that this reanalysis further confirms the ability of the capacitance clamp to effectively modify capacitance in real neurons.

Measured gain dependence on near capacitance in DGGCs: In the DGGCs, we explored a range of 60% to 300% of the original near capacitance. As the average near capacitance was ~20 pF, the total capacitance of ~120pF (near+far) thus changed in a range from ~110 pF to ~160 pF, i.e., roughly a 1.5 fold change and not a 5-fold change as for the near capacitance. To clarify this smaller relative change with respect to the total capacitance, we mention this range explicitly in the revised manuscript (lines 206-209) and show the total capacitance in the exemplary cell in Figure 4 A. As a measure of gain dependence, we performed a linear regression of gain versus absolute capacitance change for each recorded DGGC, which yielded a gain reduction of -0.10 (-0.13 to -0.06) Hz/pA^1/2^ per 10 pF (median and interquartile range for 9 cells, see lines 239-240). Gain reduction in DGGCs and the simulated neuron thus differ by a factor of ~0.5 (see next section) – a more similar effect size in experiment and model than suggested by the comparison in terms relative near capacitances.

Comparison of DGGCs gain dependence to the Wang-Buzsáki model: The revised manuscript explicitly shows how the gain of the Wang-Buzsaki (WB) neuron model depends on capacitance (see Figure 2 E and updated Table 1). At the original capacitance, the gain of the WB model is 3.8 Hz/pA^1/2^, which is about two-fold higher than the gain of the unclamped DGGCs 1.76 (1.72 to 1.86) Hz/pA^1/2^. Across the large capacitance range tested in the model from 90 pF to 210 pF, the gain roughly follows a 1/C behavior with a steep slope for small capacitances and flatter one at large capacitances. The comparison to the experiment thus depends on the capacitance range over which this reduction is evaluated. One fair way of comparison is to pick a range around the original capacitance of 150 pF corresponding to the experimentally accessible manipulations (-20 to +40 pF). In this range, the WB model gain reduction was -0.22 Hz/pA^1/2^ per 10 pF. The gain of the DGGCs thus is less sensitive (-0.10 Hz/pA^1/2^ per 10 pF), but this reduction to 50% is a much milder difference and can be partially accounted for by the different excitability of the two neuron types (see lines 240-250).

Expected dependence of gain on (near) capacitance in DGGCs: What is the origin of the smaller gain reduction in the DGGCs compared to the model neuron? One difference between WB model and DGGC lies in the composition of ionic currents that together with the baseline capacitance determine the original gain of a neuron. From this original gain, a rough expectation for the gain reduction can be formulated assuming a scaling of gain with 1/C – a good approximation for class 1 single-compartment neurons under certain conditions (Izhikevich, 2006, see added Methods section “Analytically expected effect of capacitance on the form of the f-I curve”, pages 25-26). Then close to the original capacitance C_c_, the gain reduction should be -gain(C_c_)/C_c_. As the model overall has a higher gain but approximately the same total capacitance as the DGGCs, it is therefore expected to also have a higher gain reduction, Concretely, according to this back-on-the-envelope calculation, the model gain reduction should be around -3.8 Hz/pA^1/2^/ 150 pF = -0.25 Hz/pA^1/2^ per 10 pF, twice as high as the DGGC expectation -1.8 Hz/pA^1/2^ / 127 pF = -0.14 Hz/pA^1/2^ per 10 pF. Concluding, the observed discrepancy between model and experiment in gain reduction is consistent with their difference in overall excitability.

In addition to the biological differences, a further technical factor potentially decreasing the measured gain dependence in the DGGCs is the space clamp problem. Location and geometry of the axon initial segment (AIS), as well as the distribution of active channels, determine how much the capacitance of the somatic compartment impacts AIS excitability (Goethals, 2020, *eLife*). As these factors might vary among cells, they could contribute to the observed variability in the impact of somatic capacitance. The observed significant gain reductions in the majority of DGGCs, together with the effects on the spike shape, however, demonstrate that clamping capacitance of the soma can effectively modify the capacitive load of the AIS in a real neuron.

2) Concerning the potential impact of capacitance clamp, and since most changes in capacitance in physiological contexts seem to be related to neuronal growth, it would have been really interesting to compare the impact of manipulating capacitance with the impact of manipulating concomitantly input resistance and capacitance, which is expected when neuronal size is changing, for instance during development. This type of comparison would also help underlining the significant contribution of changes in capacitance, which have been so far undermined. Testing these two manipulations in parallel would greatly help to disentangle the specific contributions of changes in membrane resistance and capacitance during neuronal growth, and would emphasize the value of the capacitance clamp tool.

We thank the reviewer for this interesting idea. Neuronal growth provides a good example of a process that cannot be studied by classic dynamic clamp alone, because a modified cell size inevitably implies an altered capacitance as well as homeostatic regulation of channel numbers. As proposed by the reviewer, an experimenter can combine the capacitance clamp with virtual conductance injection via classic dynamic clamp to map neuronal firing in the space of capacitance and leak conductance, effectively simulating the passive electrical changes associated with different growth scenarios.

Such a combined experiment is unfortunately beyond the scope of this introductory study of the capacitance clamp. But we included a section in the Discussion to emphasize this possible combination of capacitance clamp and classic dynamic clamp in the context of neuronal growth (lines 366-372).

Reviewer #3 (Recommendations for the authors):The manuscript is very clearly written and well-focused.Lines 93-94: "…change with respect to the original capacitance (e.g. Ct=67.4 pF: C=67.5 pF; Ct=336.9 pF: C=338.1 pF), whereas the… Why are these examples chosen and not a more uniform range?

Throughout the paper we used the range explored in the experiments from 60% to 300% of the capacitance to be modified and the indicated absolute capacitance values represent the corresponding limits of the tested range. As mentioned in the Discussion, this was the range where we had a reliable stable capacitance clamp in experiment.

Section 2.5.1: I found this very confusing. In an RC circuit the response sizes to a sequence of current pulses as measured by baseline to peak are identical. It is very confusing to say "…the cell's response to the second one should be higher than to the first one." Or to say "…as apparent by the larger step sizes in the stair-like voltage response and the finally higher ratio of last to first pulse response." Please rewrite.

It is correct that the response size is the same if the length of a single pulse is much longer than the membrane time constant, so that the circuit reaches its stationary state. For short pulses, however, as considered here in the context of temporal integration, the response size of the later pulses increases due to temporal summation (see Figure 5 A). We rewrote the section clarifying the context of brief input pulses accordingly (lines 259-270).

Section 2.5.3: This section was the only part of the paper I found unconvincing. It is a foregone conclusion that the CapClamp can find the critical capacitance in the Wang-Buzsáki neuron, given Section 2.3. The failure of the technique to find the critical capacitance in a dentate gyrus granule cell is thus an ambiguous result. Is this a technical failure or a real result? I suggest deleting this section.

One of our original motivations to develop the capacitance clamp was to extend the experimental tools available to systematically scan real neurons for interesting bifurcations. As some bifurcations like the presented saddle-node loop bifurcation involve fast dynamics (switch to the strongly attracting manifold, see Hesse, 2017), we thus wanted to ensure that dynamics in this regime can still be mimicked with the CapClamp. We agree, however, with the comment of the Reviewer that the presented simulation results in Figure 5 G and H (original manuscript) are difficult to interpret biologically and the interpretation of the absence of the critical transition in experiments remains ambiguous. We therefore followed the suggestion to delete this section yet took the liberty to include a few sentences in the Discussion outlining the option to detect dynamical switches with CapClamp (see lines 378-392).